# Towards long-term records of rain-on-snow events across the Arctic from satellite data

Annett Bartsch[1], Helena Bergstedt[1], Georg Pointner[1], Xaver Muri[1], Kimmo Rautiainen[2], Leena Leppänen[2,3], Kyle Joly[4], Aleksandr Sokolov[5], Pavel Orekhov[5], Dorothee Ehrich[6], and Eeva Mariatta Soininen[6]

[1]b.geos GmbH, Industriestrasse 1, 2100 Korneuburg, Austria
[2]Finnish Meteorological Institute, Earth Observation Research, P.O.Box 503, FI-00101 Helsinki, Finland
[3]Arctic Centre, University of Lapland, Pohjoisranta 4, 96101 Rovaniemi, Finland
[4]National Park Service, Gates of the Arctic National Park and Preserve, Arctic Inventory and Monitoring Program, Fairbanks, AK 99709
[5]Arctic Research Station, Institute of Plant and Animal Ecology, Ural Branch, Russian Academy of Sciences. Zelenaya Gorka Str., 21, Labytnangi, Yamal-Nenets Autonomous District, Russia
[6]UiT - The Arctic University of Norway, 9037 Tromsø, Norway

**Correspondence:** Annett Bartsch (annett.bartsch@bgeos.com)

**Abstract.** Rain-on-Snow (ROS) events occur across many regions of the terrestrial Arctic in mid-winter. Snow pack properties are changing and in extreme cases ice layers form which affect wildlife, vegetation and soils beyond the duration of the event. Specifically, satellite microwave observations have been shown to provide insight into known events. Only Ku-band radar (scatterometer) has been applied so far across the entire Arctic. Data availability at this frequency is limited, however. The utility of other frequencies from passive and active systems needs to be explored to develop a concept for long-term monitoring. The latter are of specific interest as they can be potentially provided at higher spatial resolution. Radar records have been shown to capture the associated snow structure change based on time series analyses. This approach is also applicable when data gaps exist and bears capabilities to evaluate the impact severity of events. Active as well as passive microwave sensors can also detect wet snow at the timing of a ROS, if an acquisition is available. Wet snow retrieval methodology is, however, rather mature, compared to identification of snow structure change which needs consideration of ambiguous scattering behaviour. C-band radar is of special interest due to good data availability including a range of nominal spatial resolution (10m-12.5km). Scatterometer as well as SAR (Synthetic Aperture Radar) data has been therefore investigated. Temperature dependence of C-band backscatter at VV polarization observable down to -40°C is identified as a major issue for ROS retrieval, but can be addressed by combination with passive microwave wet snow indicators (demonstrated for Metop ASCAT and SMOS). Results were compared to in situ observations (snowpit records, caribou migration data) and Ku-band products. Ice crusts were found in the snow pack after detected events (overall accuracy 82%). The more crusts (events) the higher the winter season backscatter increase at C-band. ROS events captured on the Yamal and Seward peninsulas have had severe impacts on reindeer and caribou, respectively, due to ice crust formation. SAR from specifically Sentinel-1 is promising regarding ice layer identification at better spatial details for all available polarizations. The fusion of multiple types of microwave satellite observations is suggested for the creation of a climate data record, but the consideration of performance differences due to spatial and temporal cover as well

as microwave frequency is crucial. Retrieval is most robust in the tundra biome, where results are comparable between sensors. Records can be used to identify extremes and to apply the results for impact studies at regional scale.

## 1 Introduction

Rain-on-snow (ROS) modifies snow properties and can lead to the formation of ice crusts which impact wildlife (incl. muskox, reindeer, fox, crows; Putkonen and Roe, 2003; Forbes et al., 2016; Sokolov et al., 2016; Ehrich et al., 2017) and vegetation (Bjerke et al., 2017) and related fluxes (Treharne et al., 2020). Water percolates through the snow pack impacting energy fluxes, warming the soil beneath (Putkonen and Roe, 2003; Westermann et al., 2011). Events occur across many regions of the Arctic in mid-winter and are sometimes documented through observations of local residents (Forbes et al., 2016). Quantitative accounts are challenging as they require measurements of liquid precipitation and snow properties at the same time (Serreze et al., 2021). snowpit data which detail layers and hardness can document events, but these are made manually at comparably only very few sites. Air temperature and general precipitation measurements from automatic weather stations usually serve as proxy instead. Such data relate to the trigger of snow structure change but do not reflect the impact. Satellite observations based on microwaves can document the event itself (wet snow) or the change in snow conditions following a ROS event (Bartsch, 2010b) as grain size increases and often ice crusts form which alter the signal-snow interaction. Such changes occur within a short time compared to general snow metamorphism and compaction throughout the winter season. Short term snow structure change can be frequently observed with satellites specifically over Scandinavia, northern European Russia and Western Siberia as well as Alaska (Bartsch, 2010b; Serreze et al., 2021). Wet snow (associated with ROS) detection with satellite data requires dense sampling intervals, as events can be rather short. The mapping of snow structure changes as a result of events instead of wet snow during an event circumvents this issue but requires the use of wavelengths which are sensitive to changes in snow properties (e.g. Tsang et al. (2022)), this means comparably short wavelength. Ku-band scatterometer data (2.3cm wavelength) has been shown to reflect snow conditions well in this context (Bartsch et al. (2010a); gridded to 10km nominal resolution) but C-band (5.6cm) acquisitions are much more common and to date operationally available. Forbes et al. (2016) document a ROS event where changes in C-band backscatter (12.5 km nominal resolution) resemble the patterns of a specific event reported through local communities in Western Siberia, southern Yamal peninsula. Events which had severe impacts for reindeer herding have occurred in this region several times in the last two decades. They are suggested to be linked to specific sea ice conditions (longer open water season, Forbes et al. (2016)).

Ku-band radar derived potential ROS events are readily available for 2000 to 2009, reflecting the availability period of data from Seawinds on QuikScat (Bartsch, 2010b). This is to date the only circumpolar satellite based record available (Serreze et al., 2021). The Indian Oceansat-2 and Scatsat-1 provide similar records for 2009 to 2013 and from 2017 onward respectively but have not been used for this purpose to date. Readily gridded backscatter data for Oceansat-2 are available through BYU (Brigham Young University, https://www.scp.byu.edu/data.html) for 2009-2015 (resolution enhanced version 2.2 km). C-band radar data from ASCAT (Advanced Scatterometer) is operationally available on multiple platforms (Metop-A to -C) since 2007 (e.g. Naeimi et al. (2012), 12.5 km gridding) and thus of high interest for recent and future monitoring. The C-band

scatterometer regional test by Forbes et al. (2016) indicated applicability of the structure change approach, but the parameter adjustment to this wavelength has neither been justified so far nor have such data been applied for circumpolar retrieval. The continuation with C-band would be beneficial as it has the advantage that similar information is also available from Synthetic Aperture Radar (SAR) with much higher spatial resolution (starting at 10m as for example accessible through the Copernicus Sentinel-1 mission). Temporal sampling of SAR is much lower, but its complementary value for documenting related snow hardness increase should be explored. Wet snow detection with C-band SAR is comparably mature (e.g. Nagler and Rott, 2000; Nagler et al., 2016) and has been investigated in the context of ROS for sites with exceptionally good data availability (Vickers et al., 2022). Temperature decrease has been, however, also found to increase C-band backscatter at VV polarization (as is operationally available from ASCAT) during frozen conditions (Bergstedt et al., 2018). But so far only the impact of air and soil temperature has been assessed. Comparisons to snow temperature as well as structure associated with ROS are still lacking. The implications when using the same methodology as for Ku-band (Bartsch et al., 2010a) for ROS detection needs to be quantified and method adaptions explored. This includes the combination with data from other sensors and wet snow detection.

Snow surface melting associated with ROS has been shown to be captured by passive microwave missions with focus on the detection of an event itself in several case studies (Serreze et al., 2021). Such records are partially of much coarser spatial resolution than the above mentioned scatterometer records but go comparably far back in time. SSMI records (25km) go back to 1979. They have been used so far only for the description of events on selected Canadian high Arctic islands (e.g. (Grenfell and Putkonen, 2008; Langlois et al., 2017)). Langlois et al. (2017) derive also potential ice crust forming events, but validation in the context of ROS is so far unavailable. AMSR-E2 provides higher spatial resolution (12.5km, available since 2002) and has been therefore applied in several cases for regional wet snow detection in ROS context in the past (Semmens et al., 2013; Dolant et al., 2016; Sokolov et al., 2016; Pan et al., 2018; Langlois et al., 2017). The wavelengths used by SSMI and AMSR-E2 are even shorter than Ku-band (0.8 – 1.6 cm versus 2.3 cm). L-band passive microwave observations such as provided by SMOS (2010-present, L-band, 23 cm, 50km) have not been used so far in this context but shown to be applicable for wet snow detection (Mousavi et al., 2021, 2022).

The overall aim of this study is to identify rain on snow events which alter snow conditions. Specifically, the advantages and disadvantages of using C-band radar are assessed. The objectives are to (1) gain better insight into recent occurrence of specific rain on snow events across the Arctic, (2) to explore the potential of radar data for building circumpolar long-term data records, (3) to investigate the impact of events on snow properties and (4) to assess the added value of L-band passive microwave observations for ROS detection. Objective (2) is enabled through the modification of a Ku-band scheme for C-band. In situ data not only encompass standard air temperature and snow depth records from an automatic weather station, as used in previous studies (Bartsch et al., 2010a; Semmens et al., 2013; Wilson et al., 2012; Sokolov et al., 2016; Pan et al., 2018), but are extended to snowpit observations (layer thickness and hardness, snow surface temperature) as well as automatic liquid precipitation measurements. For the first time also L-band passive microwave observations are considered in this context. The added value of the combination of passive and active microwave records is investigated. One previously documented event on Yamal (2013) is revisited and two recent ROS series are analysed in detail with C-band scatterometer as well as L-band passive

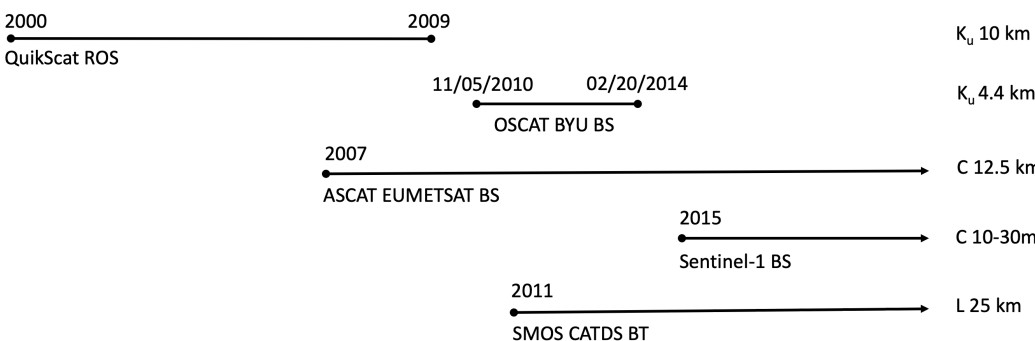

**Figure 1.** Availability of continuous records from used microwave satellite data, band ($K_u$, C and L) and nominal resolution. ROS - rain-on-snow product (active microwave), BS - backscatter (active microwave), BT - brightness temperature (passive microwave), BYU - Grigham Young University, CATDS - Centre Aval de Traitement des Données SMOS.

microwave observations (Yamal and Alaska in 2020 and 2021 respectively). The series of ROS which occurred over Yamal in late 2020 is documented through a dedicated in situ campaign and in addition analysed with C-band SAR. The potential to provide a measure of severity of a ROS event for wildlife is explored in addition based on caribou migration patterns in Alaska in case of the December 2021 events.

## 2 Satellite Microwave data

### 2.1 Ku-band scatterometer

The Quikscat (Ku-band) product of potential ROS events (Bartsch, 2010a) is based on HH polarization acquisitions and starts with winter (November to February) 2000/01 and ends winter 2008/09 (Figure 1) due to instrument failure. The pencil beam antenna design allowed for several measurements per day (Bartsch et al., 2010c) compared to Metop ASCAT (C-band) which provides 80% global coverage only every day in case of one satellite. Oceansat-2 (also referred to as OSCAT) theoretically provides continuity for Quikscat, but the processing used for the ROS product is based on a different scheme (Technical University Vienna) than publicly available for OSCAT (through BYU - Brigham Young University). The type of gridding (full footprint resampling versus partial footprint resampling) differs, leading also to different nominal resolution. A transfer of the QuikScat approach has been nevertheless tested in order to investigate the potential for the production of a longer data record. The so called 'egg' product has been selected (4.4km nominal resolution, Long (2017)). The Oceansat-2 (OSCAT BYU) product starts 5th November 2010 and ends 20th February 2014. The first and last winter (as defined for the QuikScat ROS product) is therefore incomplete. HH polarization is in both cases, QuikScat and Oceansat-2, available from the inner beam acquisitions. Incidence angles differ slightly with 46° and 48.6° respectively.

## 2.2 C-band scatterometer and Synthetic Aperture Radar

The Advanced Scatterometer (ASCAT) instrument onboard the Metop satellites provides C-band (5.2 GHz) VV-polarized backscatter data sets at a spatial resolution of 25–50 km (Figa-Saldaña et al., 2002). Metop A, B, and C have been launched 2006, 2012 and 2018 respectively. The first records suitable for use start 2007. The data provided by EUMETSAT come in a hexagonal grid (nominal resolution 12.5 km) as part of the soil moisture data product (Naeimi et al., 2009). The provided backscatter value is normalized to a 40° incidence angle. Data are continuously available since 2007, providing global coverage with a temporal resolution of approximately daily observations.

The two earth observation satellites Sentinel-1A (launched in April 2014) and Sentinel-1B (launched in April 2016) have an identical C-band SAR sensor onboard (Schubert et al., 2017). Most land area is monitored with Interferometric Wide Swath (IW) mode (10 m nominal resolution). Near coastal Arctic tundra regions are, however, only covered in Enhanced Wide Swath (EW) mode in medium resolution (40 m) due to requirements for sea ice monitoring. This is specifically the case for the Yamal peninsula, a focus area of this study. Information can be captured in dual polarization (HH+HV or VV+VH; H—horizontal, V—vertical). Mostly HH+HV is available for EW mode in high latitude areas, including Yamal. VV+VH is acquired in IW mode for Scandinavia. GRD (Ground range detected) products were used for this study. GRD products are detected, multi-looked, and projected to ground range using an Earth ellipsoid model (Potin et al., 2014; Potin, 2013).

## 2.3 L-band radiometer

The SMOS (Soil Moisture and Ocean Salinity) satellite payload instrument MIRAS (Microwave Imaging Radiometer with Aperture Synthesis) observes the Earth at L-band, specifically at bandwidth reserved for radio astronomy (1404 MHz – 1423 MHz). The instrument uses two-dimensional aperture synthesis to provide a series of snapshot images reconstructed from simultaneous measurements of multiple receivers. SMOS observes the target areas at wide incidence angle range (0-60 degrees) and provides fully polarized brightness temperatures at a spatial resolution of 35 to 50 km depending on the measurement geometry. SMOS started the continuous observations in 2010, record use for this study starts however only winter 2011/12 due to data gaps during the first winter. Data used here are from Centre Aval de Traitement des Données SMOS (CATDS), the French ground segment for the SMOS Level 3 and 4 data. CATDS level 3 daily brightness temperatures (L3TB) form the basis for this study. CATDS data are processed from the SMOS L1B products from ESA. The data are binned and averaged into fixed incidence angle classes at five degrees intervals from 0-5 degrees to 60-65 degrees and available on the EASE2 (Equal-Area Scalable Earth) grid projections. The CATDS product includes flags for potential man-made radio frequency interferences (RFI) and Sun contamination for users to discard data not suitable for their use.

The vertically and horizontally polarized brightness temperature data are used at the incidence angle range from 45 to 50 degrees in this study. Due to the measurement geometry, there are typically most data available at incidence angles between 25 to 40 degrees. However, at 50 degrees incidence angle the effect of a dry snow cover to the observed signal at vertical polarization is at minimum (see section 4.1, Schwank et al. (2015); Lemmetyinen et al. (2016)). The selected incidence angle

range was a trade-off, it provided a good daily coverage and was close to the optimal incidence angle for L-band observations over the snow areas.

## 3 In situ observations and auxiliary data

In situ data comparison and the discussion of specific rain-on-snow events focuses on northern Scandinavia (Finland and Norway), the Yamal Peninsula (Russia) and Seward Peninsula (Alaska). The Sodankylä site in Finland is characterized by boreal forest and all others mostly represent shrub tundra (Figure 2, #1).

### 3.1 Automatic measurements of meteorologic conditions in northern Finland

Automatic recordings at the Sodankylä experimental site (intensive observation area) include standard AWS data (including air temperature and snow depth) as well as liquid precipitation measurements (referred to as LRI in the following) observed with a Laser Precipitation Monitor (Thies). All these data types are available for the entire length of the analyses period for Metop ASCAT. Daily values are used for this study.

### 3.2 Snowpit observations

Three types of snowpit records have been available for the study: weekly, once per season and campaign data. They represent a gradient in northern Scandinavia (#1-3 in Figure 2) and on the Yamal peninsula (#4 in Figure 2).

Of specific interest are snow hardness (Table 1) observations as they reflect presence of ice crusts which can be caused by rain on snow events. The thickness of distinct layers is recorded in addition to a hardness class. The same scheme is used for the sites on Varanger, Saariselkä and Sodankylä.

Regular, mostly weekly observations are made at Sodankylä, as part of an experimental site (intensive observation area - IOA) in 2009-2018 (Leppänen et al., 2016). Since 2018, the snowpit measurement site was relocated to a forested area nearby due to changes in experimental configuration. The IOA site is located within a 40m wide forest clearing, but the associated ASCAT footprint covers peatland to a large extent. A second snow monitoring site is, however, located within a peatland (referred to as BOG in the following). For BOG, snow hardness measurements are not available from all years and at less regular intervals than at IOA, but snow temperature and snow water equivalent (SWE) has been measured in case of most observation dates. The more representative peatland site is therefore used to evaluate C-band backscatter with respect to temperature and SWE. The IOA snow hardness data serve as evaluation of detected events.

Several long-term monitoring sites by Climate-ecological Observatory for Arctic Tundra (COAT, University of Tromsø) are located on the Varanger peninsula. snowpit information has been collected every March since 2006. The same snow hardness classification as used at Sodankylä (see Table 1) has been used since 2017. 66 snowpit locations within 6 ASCAT grid points are available (up to 18 per point, 6 on average). The sites represent different types of vegetation (sites with and without tall shrubs). A further site has been established recently near Saariselkä (Niilanpää) by FMI (Finnish meteorological Institute). This site is located between Varanger and Sodankylä and has been previously used for L-band passive microwave analyses

**Table 1.** Overview of terms for snow hardness documentation for available snowpit observations

| Site (s) | Term | Hand hardness test | Number | considered for evaluation |
|---|---|---|---|---|
| Sodankylä, Saariselkä, Varanger | Very soft | Fist | 1 | No |
| | Soft | 4 fingers | 2 | No |
| | Medium | 1 finger | 3 | No |
| | Hard | Pencil | 4 | Yes |
| | Very hard | Knife blade | 5 | Yes |
| | Ice | | 6 | Yes |
| Yamal | Firn | Pencil or knife blade | | Yes |
| | Ice | | | Yes |

(Derksen et al., 2017). The first snowpits have been made in 2021. It represents tundra but is located close to the treeline. The overlapping ASCAT footprint includes both, tundra and forest.

A snowpit transect of almost 400 km length with samples about every 20 km has been made on the Yamal peninsula, Western Siberia, in February 2021. One of the aims was to document the impact of a series of rain-on-snow events in early winter 2020. Five snowpits have been made per site (one in centre of a square). Crust properties (hardness and thickness) have been measured at all five points. Two crust classes are considered for evaluation (Table 1), firn and ice. Firn crusts consist of frozen individual conglomerates of snow and ice (degree of hardness - pencil or knife test; Table 1). Density was recorded for the central point and is available for 16 of the 19 sites along the transect.

### 3.3 GPS collar collection for caribou movements in Alaska

From September 2009 to April 2021, female caribou from the Western Arctic Herd, in northwest Alaska (#5 in Figure 2), were equipped with GPS collars by the Alaska Department of Fish and Game and the National Park Service. All capture procedures were approved by the State of Alaska Institutional Animal Care and Use Committee (permits 2012-031R and 0040-2017-40). From 2009-2019, captures were performed as the caribou crossed the Kobuk River during their fall migration. From 2019-2021, captures were performed using helicopter-based net gunning procedures. Caribou exhibit their lowest movements rates during mid-winter and are relatively sedentary during this season (Joly et al., 2020).

### 3.4 Further external records

ERA5 reanalyses data (air temperature) have been used for the documentation of events across Yamal as no in situ meteorological records have been available along the snowpit transect. The data was available from the European Centre for Medium-Range Weather Forecasts (ECMWF) and accessed via the Climate Data Store (CDS). Daily maximum, minimum and mean values of 2-metre air temperature were derived from the hourly available 2-metre ERA5 air temperature data.

Several ROS events documented by local observers are listed in Pan et al. (2018) for Fairbanks, Alaska (64.80°N, 147.88°W). In total eight events fall within the analyses period of ASCAT. The records are used to discuss the precision of the date which is automatically extracted.

Landcover information which includes lakes (Lamarche et al., 2017) has been used to consistently mask all satellite records. The boundary of the Circum Arctic Vegetation Map (CAVM, Raynolds et al. (2019)) has been used for the separation of tundra for the interpretation of the results. Sea ice concentration records are used for visualisation and discussion of specific events. They are based on AMSR-E provided through University of Bremen based on Spreen et al. (2008).

## 4    Methods

### 4.1    Considerations for C- and L-band use

Microwave interaction at the landsurface is impacted by surface roughness, volume scattering and dielectric properties (e.g. Ulaby et al., 1986; Woodhouse, 2006). Snow and soil properties vary throughout the winter due to structure and phase change which is reflected to a certain degree by the wavelength used (e.g. Bartsch, 2010c). When a ROS occurs, the snow pack is dry before the event, turns wet during and right after the event until the snow pack refreezes and becomes 'dry' again. Wet versus dry detection is a well explored and common application of microwave observations from satellites (e.g. Nagler and Rott (2000) for C-band and Bartsch et al. (2010c) for Ku-band). When the liquid water content increases in the snowpack, radar backscatter decreases significantly. For L-band passive sensors, such as SMOS payload, the observed brightness temperature at horizontal polarization is significantly decreasing with increasing liquid water content. The vertical polarization signal is not affected as much and may even increase at certain conditions (Pellarin et al., 2016). Wet snow is eventually detected via change detection using thresholds applied to backscatter/brightness of single bands or polarization ratios. The focus of such detection schemes is, however, usually on spring melt. ROS eventually alters the structure of the snow as grain size increases and the formation of ice layers allows the detection of event occurrence based on subsequent backscatter change at Ku-band (Bartsch et al., 2010a).

The actual magnitude of backscatter change associated with the snow structure change has not been analysed so far as it depends on snow pack history and general landscape properties influencing volume scattering and surface roughness. Ku-band backscatter is also influenced by the amount of snow, the more snow the higher the backscatter (Bartsch et al., 2007). This effect is less present for C-band as the signal penetrates through the snow pack when ice layers are absent (Naeimi et al., 2012). The underlying soil properties may have an effect in this case. When liquid water content is increasing in soils, backscatter increases. Surface roughness also increases the return signal. Roughness may change for example due to tillage, but across the Arctic agriculture is largely absent, so the assumption can be made that roughness stays constant from year to year, but large spatial variations can be observed across different landscapes. This roughness component needs to be considered when absolute temporal variations from different sites are compared with each other. In autumn when the ground freezes, backscatter decreases specifically for C-band due to the drop in dielectric constant (e.g. Naeimi et al. (2012)). For scatterometer data this can be observed to take several weeks as topography within the footprint may vary (Bergstedt et al., 2020). When freeze-

up is complete, roughness and volume scattering can be assumed to govern backscatter response. If any other effects are absent, early winter can be used as a representation for surface roughness (Bartsch et al., 2016). There are, however, also other mechanisms than snow structure change which lead to higher values at C-band. Comparisons with soil and air temperature records under frozen conditions suggest a dependence of C-band backscatter at VV polarization (Bergstedt et al., 2020). The lower the temperature the higher the backscatter. This has been found to be independent from the amount of overlying snow. A sudden drop in temperature (with respect to a chosen time window used in change detection) may therefore show a similar magnitude like the formation of ice layers. So far, this effect has been analysed using absolute backscatter (reflecting roughness, structure/volume and dielectric properties together). To exclude the roughness component, and enable spatial comparability, the location specific surface roughness needs to be determined based on early winter backscatter statistics (minimum) in a pre-processing step. The temperature effect can be then analysed by comparison to in situ snow properties (surface temperature, snow water equivalent).

At L-band the relative permittivity of free liquid water is very high, the real part is close to 90 (Maetzler, 2006). This enables many applications, such as detection of liquid water content of soil, and monitoring of soil freeze and thaw cycles. The same applies for detection of ROS events. The emissivity from the frozen soil at L-band is high for both vertical and horizontal polarisation due to the low permittivity (Hallikainen et al., 1986; Maetzler, 2006). The attenuation by a dry snow layer on top of the frozen soil is negligible at L-band. A ROS event creates a layer of wet snow on top of the snow cover possibly with some degree of free liquid water within the snow layer. The free liquid water and wet snow decreases the emitted power and thus the measured brightness temperature $Tb$. The incidence angles around 50 degrees are specifically good for detection of ROS events. Even though a dry snow layer does not attenuate the emission from the frozen soil, it affects the measurements at L-band as a result of impedance matching and changes in the refraction angle at the snow–soil interface (Schwank et al., 2015; Lemmetyinen et al., 2016). At the incidence angles around 50 degrees, the effect is minimal at vertical polarization, while for horizontal polarization the brightness temperature is increased due to the dry snow cover. When soil is frozen and covered with dry snow the normalised polarisation ratio, $NPR = (Tb_{vpol} - Tb_{hpol})/(Tb_{vpol} + Tb_{hpol})$ reaches its minimum value. The wet snow layer and the introduction of free liquid water within the snow layer increases the NPR value. It is therefore possible to set an empirically defined threshold for detection of possible ROS events.

Passive microwave observations commonly provide two polarizations for a certain frequency and are taken at a range of incidence angles. Band ratios are commonly used for wet snow applications. In case of SSMI/SSMR the channel difference between 37V-19V (or 18V) (SSMI/SSMR) is used and in case of L-band (SMOS and SMAP) the normalized polarization ratio (Mousavi et al., 2021, 2022), since wet snow affects the horizontal polarization increasing the V-H difference significantly. The NPR magnitude under dry snow conditions varies spatially. Basic statistics (winter average and standard deviation) which describe dry snow conditions for each grid cell need to be therefore considered for threshold determination (Mousavi et al., 2022).

## 4.2 Pre-processing of satellite data

ASCAT data available through EUMETSAT as gridded products with 12.5km grid spacing were rearranged into time series for all land area grid points north of 50°N. Data is provided as $\sigma^0$ values (in dB) normalized to an incidence angle of 40°. The ASCAT gridding as described in Naeimi et al. (2009) is used. A unique ID is assigned to each grid point (referred to as GPI) of the hexagonal representation. A daily temporal resolution is chosen although data gaps can occur towards the south. A (dry/frozen) reference layer has been derived in a first step based on November backscatter (minimum of entire record) for each grid point (Figure 2). The difference to the actual backscatter is then derived and in the following referred to as $\Delta\sigma^0$.

The Oceansat-2 'egg' dataset has been processed similarly to the QuikScat product as described in Bartsch et al. (2010a). This includes the use of HH polarization (inner beam) as well as a static threshold of 1.5 dB for snow structure change identification.

The full polarized SMOS L3 brightness temperature data are available from CATDS in a 25km regular EASE2 polar grid with daily maps (Bitar et al., 2017). This study uses the horizontal and vertical polarized brightness temperatures at incidence angle bin from 45 degrees to 50 degrees. Bad quality data has been masked out based on the standard deviation and accuracy information given in CATDS data for the pixel-wise brightness temperatures and based on the amount of observations that are suspected to be contaminated by RFI. The Normalized Polarization Ratio (NPR) was determined from the masked data similarly as in SMOS and SMAP freeze and thaw products (Rautiainen et al., 2016; Derksen et al., 2017). In a further masking step, sites with NPR standard deviation larger than 0.02 have been excluded from further analyses. A relatively high standard deviation occurs along coastlines and further regions with high RFI not captured in the CATDS masking and is considered in a later step.

## 4.3 ROS retrieval

ROS related snow structure changes within the snow pack can be identified by time series analyses of radar backscatter (Bartsch, 2010b). The averaged backscatter of the three days before a specific day are compared with the three days after (dry/frozen conditions in both cases). A certain threshold of backscatter increase needs to be exceeded. So far this threshold has been determined based on single observed events and defined as static (Bartsch et al., 2010a; Forbes et al., 2016), as one value for the entire Arctic. The threshold varies between sensors due to differences in wavelength and noise characteristics. In case of QuikScat (Ku-band) 1.5 dB and for ASCAT (C-band) 0.5 dB have been used. Bartsch et al. (2007) suggest the consideration of location specific noise in case of snow related time series analyses of Ku-band (three times the standard deviation for detection of snow melt). This has been so far not tested for C-band. In general, the noise of the used C-band scatterometer data type (fan beam) is much lower than for Ku-band (pencil beam; Bartsch (2010b); Bartsch et al. (2010c)). A threshold of one standard deviation is therefore used. The location specific standard deviation has been obtained from November to February records (Figure 2).

Some regions are characterized by very low standard deviation (e.g. most of eastern Siberia) due to the general aridity and cold temperatures in winter (low snow depth and continuous frozen conditions). The overall difference between the average

standard deviation and its standard deviation are considered for this purpose. The resulting value (0.2 dB for $\Delta\sigma^0$) serves as a minimum threshold. The three-day window approach is then applied to the masked dataset to identify sudden change in snow structure (Table 2).

Snow surface temperature measurements from snowpits were used in addition to further analyse the relationship with backscatter as error source. Detected events were eventually compared with weather station records of liquid precipitation (LRI) at Sodankylä in order to investigate false detection due to temperature change.

## 4.4 Combination with SMOS wet snow retrieval

Eventually, C-Band time series were compared with L-band passive microwave records in order to assess the added value for temperature error correction in the C-band retrievals. NPR increases when air temperature rises above zero degree Celsius, indicating wet snow. NPR level and noise varies from location to location (examples shown for SMOS in Figures 3, 4). This needs to be considered for wet snow detection. Similarly to ASCAT ROS retrieval, a location (grid point) specific threshold has been defined based on the long-term mean (location specific) and standard deviation (Table 2). Wet snow is assumed only when the NPR exceeds the average plus the standard deviation multiplied by a factor. Mousavi et al. (2022) suggest a factor of 5 for ice sheets. The higher the factor the more events might be omitted. A factor of 3 is applied for seasonal snow based on observations at the Sodankylä reference site (e.g. early November event in 2012, Figure 3). The second step of brightness temperature threshold application as suggested by Mousavi et al. (2022) is omitted as wet snow detection is not the primary indicator but only used in a second step for ASCAT event confirmation.

ASCAT ROS results are eventually tested for wet snow occurrence in SMOS to separate ROS from cold spells. Grid point centres are considered for fusion of the datasets. The closest point is used in each case. A time window of $\pm$ 3 days with data is applied in addition in order to account for timing mismatches as well as SMOS data coverage issues. The temporal precision is expected to be lower for ASCAT ROS than for SMOS as the algorithm by Bartsch (2010b) is based on the assumption that events occur on a single day. When events last longer (e.g. Figure 3), the determined date from ASCAT depends also on the magnitude of backscatter change and duration. For the 2013 event on Yamal, two days in a row are detected with wet snow. ASCAT ROS is identified for the day after the second day (Figure 4).

## 4.5 Pre-processing for evaluation and product cross-comparison

Results are evaluated using snowpit information from specifically the Yamal transect (#4 in Figure 2; one winter) and repeat surveys from Northern Norway and Northern Finland ((#1-3 in Figure 2). In general the presence and thickness of ice layers is used as indicator for a preceding event. Layer thickness for hardness 4, 5 and 6 (see Table 1) has been extracted from the records. Sufficient samples for accuracy quantification including error of omission and commission are only possible for Scandinavia. In order to account for variations within the ASCAT footprint, observations from Varanger and Saariselkä (Niilanpää) have been averaged for sites within the same grid point (based on cumulative thickness for each hardness class; only for grid points with at least two sites in a specific year). In case of the Yamal transect, each site is located within a different grid point. The

profile with maximum number of layers among the five snowpits per site was derived as well as the average cumulative layer thickness for the firn and ice classes.

We used caribou locations from January 2010-2017 and January 2022 and attributed each location with the value of the cumulative $\Delta\sigma^0$ of events in December 2021. In essence, we compared the intensity of icing experienced by caribou in January 2022 to the intensity of icing they would have experienced at that time if they were located where they were found in years

when they migrated to the Seward Peninsula previously (2010-2017). Caribou distributions were generalized by creating Kernel Density Estimates (KDEs) for January 2022, as well as for January 2010-2017 combined, for visualization. The 95% isopleths were derived.

To enable satellite product cross-comparison QuikScat, OSCAT and ASCAT/SMOS results were aggregated for specific winters (November to February). For comparison all datasets have been resampled to the 10x10km grid previously used for

QuikScat ROS detection (see Figure 1). The sum of all events is divided by the number of grid points. Consistent masking for water and glaciated area is used. Greenland is excluded as it is not covered in the QuikScat product. Locations with high RFI in SMOS are also excluded from all products.

## 4.6   Assessment of polarization differences at C-band using SAR

In order to test further polarizations at C-band, Sentinel-1 HH+HV (Yamal) and VV+VH (Sodankylä) data have been analysed.

Due to the much lower revisit intervals of Sentinel-1 compared to Metop ASCAT, an averaging over a 3-day window is not feasible. The SAR scenes are therefore analysed only in case of detected events with ASCAT and not for event detection itself. These synthetic aperture radar data (SAR) have been preprocessed as outlined in Bartsch et al. (2020), the backscatter coefficient $\sigma^0$ is derived and normalized to an incidence angle of 40° for comparability with ASCAT. Late October 2020, after freeze-up, is considered for the retrieval of the frozen reference. This reference is compared to acquisitions from December

2020 in case of Yamal, right after the last potential ROS event before temperatures dropped below -10°C in order to account for temperature effects (Bergstedt et al., 2020). In case of Sodankylä, only one scene before and one after a specific event in 2016 was processed (only pair in Sentinel-1 archive with snowpit data as well as event in between). Results have been re-sampled over a 100m window in order to account for SAR specific noise (speckle). With respect to the high heterogeneity of Arctic landscapes (leading to variations of speckle patterns) and the change of nominal sampling itself, the average was calculated

rather than a speckle filter applied.

## 5   Results

## 5.1   ASCAT $\sigma^0$ and SMOS NPR statistics

C-band backscatter for frozen soil conditions obtained from the beginning of the winter season (frozen reference; minimum derived from all years) varies across the Arctic (Figure 2). The pattern reflects mostly terrain but also vegetation with low

values in tundra lowlands (e.g. location #4, around 20dB) and higher values in all mountain areas and forested regions. The

standard deviation for mid-winter (November to February) backscatter is less than 0.5 dB for most of the Arctic and on average 0.38 dB. It is higher over forested areas and in regions with a high open water fraction such as parts of the Alaskan North Slope. The sites with snowpit records represent a gradient for the frozen reference and standard deviation. The Sodankylä test site is surrounded by forest and therefore shows medium variation with standard deviation of 0.4 dB as well as a relatively high
reference backscatter of $\sigma^0$ -13.6dB. The standard deviation is similar for all other in situ and ROS case study regions, but the frozen reference is lower for the Yamal and Varanger peninsulas.

Typical SMOS NPR values are within the range of zero to 0.12, exceeding 0.04 to 0.06 when the snow is wet (Figure 3). The standard deviation is approximately 0.009 at high latitudes (>60N) for the winter months (Table 2).

## 5.2  ASCAT $\Delta\sigma^0$ sensitivity to snow properties under dry snow and none-ice layer conditions

Only dates with negative temperature and snow hardness H<5 at the BOG site (Sodankylä) were considered in this context. Conditions beyond these thresholds are expected to represent different dominant backscatter mechanisms. ASCAT C-VV backscatter increased up to 4.6 dB above the long-term November minimum between November and February for the dates with snowpit observations (Figure 6). $\Delta\sigma^0$ values ranged mostly between 1dB and 4dB.

In general, 39% of $\Delta\sigma^0$ variation can be explained through temperature (note, a 12.5 km grid is compared to an in situ point
measurement, so $R^2$ might be even higher). A backscatter increase of about 2-3dB can be expected with a snow surface temperature drop down to -25°C. The comparison with snow water equivalent confirms the low influence (compared to temperature, $R^2$ 0.08) of changes in snow amount on C-VV backscatter.

## 5.3  Event detection

As an example, a time series of ASCAT, SMOS and the automatic weather station data (air temperature and liquid precipita-
370 tion) at Sodankylä are shown for winters 2012/13 and 2016/17 in Figure 3. SMOS NPR increases when liquid precipitation is recorded as for mid-November 2016. But not all occasions with liquid precipitation resulted in an increase of NPR or backscatter, as for example in mid-December 2016.

The majority of events detected by ASCAT at this site are related to short term temperature change and have been correctly masked in the second processing step (Table 2) based on SMOS NPR. Such false detection occurred each winter and outweigh
correct identification cases.

The number of potential events per grid point for the entire Arctic dropped by about 89% through combination with SMOS data (Table 2). This agrees with observations at Sodankylä. For example, only one out of five of the events could be confirmed with SMOS NPR (Figure 3) at the AWS site in November to February 2012/13. The erroneous detection due to temperature drops can be confirmed with AWS air temperature data in all cases.
The general spatial pattern of event occurrence agrees between QuikScat, Oceansat-2 and ASCAT/SMOS results north of 65°N (Figure 7). Northern Europe to Western Siberia experience frequent events as well as coastal regions along the Bering Sea affecting Alaska as well as Far East Siberia. The magnitude, however, differs between the products. The Ku-band based results (QuikScat as well as Oceansat-2) show more events than the ASCAT/SMOS accounts. Deviations occur specifically

**Table 2.** Processing steps and results for potential Rain-On-Snow (ROS) identification by fusion of ASCAT backscatter and SMOS Normalized Polarization Ratio (NPR). Events represent November – February, years 2011-2022.

| Processing step | Input | Time window | Threshold definition | #events >65°N |
|---|---|---|---|---|
| backscatter anomalies based on ASCAT (potential ROS) | backscatter | three days before versus three days after | difference >location specific winter standard deviation, but at least exceeding overall standard deviation (0.2dB) | 3068606 |
| wet snow masking with SMOS | potential ROS dates ASCAT, SMOS NPR | within three days before and three days after | NPR > local mean + 3x winter overall standard deviation (0.009) | 332327 (11%) |

over southern forested regions. When compared over tundra, average results are similar between the QuikScat and the AS-CAT/SMOS results. This includes the magnitude of events as well as the variability from year to year. Approximately 0.31 and 0.27 events per grid cell and year have been identified with QuikScat and ASCAT/SMOS respectively (standard deviation 0.06 and 0.04). The standard deviation of events is highest over Scandinavia, Western Siberia and Western Alaska (Figure 8). November is in general the month with most events for mid-winter, but December and January is also common for Alaska.

### 5.4 Characteristics of specific known documented events

#### 5.4.1 Snowpit observations Sodankylä (2012-2019), Varanger (2017-2019) and at Saariselkä (2021)

Snowpit observations have been compared to the final product (masked results) for cases where potential events have been confirmed by wet snow observations. In situ data collected at Sodankylä allow for analyses of specific months for most of the ASCAT period. Detected events at this location occurred specifically in November. The thickness of distinct layers with hardness 4 and/or 5 have been more than 10 cm in years with events (2012 and 2016) (Figure 10). Liquid precipitation has been recorded for the periods of the events in both years (Figure 3). ASCAT C-VV backscatter increased by more than 1 dB for the first events in the season although only little snow was present in both years. Sentinel-1 VV/VH was available before and after the November 2016 event (15.11. and 27.11). Sentinel-1 C-VV backscatter increased more than for ASCAT, by 4.3 dB and in case of VH by 2.1 dB.

The average March cumulative thickness of layers of hardness 4, 5 and 6 varies from year to year on Varanger (Figure 12). It was lowest for 2018 and 2020, the two years without events at this site. Only some thin layers with hardness 4 and 5 could be observed at Niilanpää in 2021, where also no event was recorded.

The highest hardness observed at Sodankylä was H5 and on Varanger H6. An overall event accuracy of above 80% can be determined when such layers occur. The overall classification accuracy as well as error of omission and commission based on the snowpit records is lowest and highest respectively when only hardness class 4 is considered as proxy for a potential ROS

**Table 3.** Event detection accuracy based on snowpit records representing early winter (Sodankylä) and late winter (Varanger and Niilanpää). Differentiation by hardness class H. Minimum cumulative thickness per hardness class 1 cm.

| Sample characteristics | Hardness class | Overall accuracy | Error of omission | Error of commission |
|---|---|---|---|---|
| Sodankylä, late November/ early December snowpits, 8 samples from 1 grid point and 8 years | H4 | 45.45 | 50.00 | 66.67 |
| | *H5* | *87.50* | *0.00* | *50.00* |
| | H4+H5 | 75.00 | 50.00 | 0.00 |
| Varanger and Niilanpää, March snowpits, 16 samples from 6 grid points and 5 years | H4 | 43.75 | 56.25 | 0.00 |
| | H5 | 62.50 | 46.15 | 0.00 |
| | *H6* | *81.25* | *25.00* | *14.29* |
| | H5+H6 | 62.50 | 46.15 | 0.00 |

(Table 3). Both type of observations (early and late winter) show that detected events may not result in hardness class 4 (pencil test, see Table 1).

### 5.4.2 Yamal 2013

A ROS event has been documented for southern Yamal, Western Siberia, in Forbes et al. (2016) with an initial analysis of ASCAT for a small spatial and temporal subset. It was also confirmed based on AMSR-E2 (Sokolov et al., 2016). No in situ measurements are available but reanalyses data and reports from reindeer herders. Reindeer herders reported that the catastrophic ROS event began on 8–9 November 2013 with about 24 h of rain, after which temperatures dropped and remained below freezing for the remainder of the autumn and throughout the winter. Wildlife impacts could be observed for foxes and crows (Sokolov et al., 2016; Ehrich et al., 2017). The event can be also confirmed through SMOS NPR (wet snow; Figure 13). Snow structure change can be detected with ASCAT already on the 7th of November, to the south of Yamal. The ROS extended further to the west on the 10th and 11th of November. SMOS and reanalyses data also confirm the detection issues by ASCAT related to temperature drops after the ROS event. Three out of four detections during winter 2013/14 can be attributed to this phenomenon. The ROS event in November was the only one in midwinter in that season.

### 5.4.3 Yamal 2020

Several documented ROS events occurred between late October and end of December in 2020 across Yamal. Events occurred with partially spatial overlap. According to the in situ measurements in February 2021, the snow height was on average 28 cm. Firn and/or ice layers were found at all sites (Figure 14).

Both, satellite and in situ observations, show that the 2020 events have affected most of the Yamal peninsula (Figures 14 and 5). The first warm period at the end of October lasted for several days. It occurred shortly after freeze up, so the starting backscatter level has been comparably low (Figure 4, -15.5 and -17 dB). $\sigma^0$ increased by about 1dB after the event in the southern part as well as the northern part of the Yamal transect (Figure 5a and 4). The November 2020 event on Yamal progressed from north to south. It is captured earlier at field point #18 (Figure 4b) than at #3 (Figure 4a). The third ROS phase

lasted from end of November to beginning of December and was largely confined to central and northern Yamal. $\sigma^0$ increased, however, also in the south but no wet snow was detected by SMOS at for example #3. Air temperature also remained at just below zero degree Celsius.

The northern part of Yamal shows the highest number of events and also $\Delta\sigma^0$ in both ASCAT VV and Sentinel-1 HH and HV (Figure 14). This section of the transect has also sites with multiple firn layers or an ice layer (north from point #14). Snow density as well as the crust thickness within the snowpack is also higher in the central part (starting from point #6). Sentinel-1 HH $\Delta\sigma^0$ increases at a similar magnitude like ASCAT VV $\Delta\sigma^0$ from south to north, from about 1 dB to 3-4 dB. Sentinel-1 HV varies more and deviates specifically in the northern part where the third event was observed. Cumulative ASCAT $\Delta\sigma^0$

decreases from event to event. It is lower than the post-event $\Delta\sigma^0$ in several cases where only one event is detected indicating that an event has been missed.

### 5.4.4   Alaska 2021/2022

The event in central and western Alaska in late 2021 had two phases. An affected area of about 500 x 500 km showed a cumulative backscatter increase of more than 1 dB. The first event started on the 17th of December and the second around the

22nd. They had partial spatial overlap. Sea ice extent declined during the events and the ice boundary retreated beyond the long-term median extent for December (Figure 15).

SMOS data coverage drops at latitudes below approximately 65°N due to the polar orbit and limited swath size (Figure 7). The Alaska event occurred mostly south of this limit what leads to failures in wet snow detection in addition to the influence of complex terrain and RFI on the SMOS NPR (Figures 15c and d).

## 6   Discussion

### 6.1   Scattering and brightness behaviour

The previously suggested static threshold for C-band backscatter increase after ROS (0.5dB, Forbes et al. (2016)) exceeds the average standard deviation of winter time backscatter in the high latitudes (0.38dB, Figure 2). In the same regions it exceeds 0.5dB (e.g. Canadian High Arctic). This may lead to false detection with the static threshold. Problematic are multiple events

as in the case of Yamal 2020. The magnitude of backscatter change decreases from event to event. The third event lead to an increase of just below 0.5dB (Figures 14 and 4). This third event was most pronounced regarding SMOS NPR (wet snow indicator). First events within the season usually reach a $\Delta\sigma^0$ of about 1.5dB (e.g. Figure 14) although little snow is present (less than 20cm, Figure 3). The observed SMOS NPR winter range agrees with values previously reported for ice sheets (Mousavi et al., 2021).

The comparison to snow surface temperature observations confirms previous analyses which had a focus on soil and air temperature (Bergstedt et al., 2018). The correlation with C-band backscatter can be expected to be more pronounced than for the Sodankylä site for most parts of the Arctic where even lower temperatures occur. The potential increase of backscatter

when temperature drops can exceed changes related to snow structure change of single events which underlines the need for a more complex approach than previously suggested for Ku-band backscatter (Bartsch et al., 2010c), a method which relies on short term backscatter increase only.

## 6.2  Validation results

Snowpit observations extend previous evaluation approaches. So far only proxies from AWS or local community observations have been used for quantitative assessment (Table 4). The accuracy results based on snow hardness class 5 (Table 3) are similar to or higher than in previous studies. 72% was reported in a case where ROS observations by observers in three settlements in Northern Quebec from one winter have been used to evaluate AMSR-E results (Dolant et al., 2016). Overall accuracy based on proxies from AWS across Alaska were 83% (QuikScat, Wilson et al. (2012)) and 86% (AMSR-E, Pan et al. (2018)). The accuracy calculations from the March snowpits might be impacted by other processes which lead to relatively hard layers. Notably wind compaction can be important in this area and result in layers with H4 (pencil test). Our results show however that layers with H5 or higher are correlated with detection based on the thresholds chosen.

## 6.3  Sensor combination and cross-comparison

The differences between the Ku-band results and the combined ASCAT/SMOS events in forested regions (Figure 2) may not be related to noise in the shorter wavelength for most of the Arctic as it is three times lower than the 1.5 dB threshold used for Ku-band (Bartsch et al., 2010b). The role of polarization as well as the frequency/wavelength need to be studied in more detail. Currently HH is used for Ku-band and VV for ASCAT. The shorter wavelength of Ku-band leads to a higher sensitivity to changes in the snow pack. More detailed analyses is required to identify the reasons for the differences as well as the correctness of the results of the Ku-band application in forested areas. This includes potential improvements by use of location specific thresholds as well as extension of validation. So far only air temperature measurements have been used for the assessment (Bartsch et al., 2010c; Serreze et al., 2021). Differences among the used Ku-band products can be also expected related to sampling methods. Results may differ for example between the BYU 'slice' and 'eggs' products. BYU also provides an ASCAT product alternative to EUMETSAT (Lindsley and Long, 2016). The 'egg' sampling to 4.45 km and normalization to 0° incidence angle may lead to differences in magnitude and noise and thus also ROS detection. Tao et al. (2022) propose that fusion of QuikScat and ASCAT backscatter records provides applicable time series for vegetation and soil analyses but the suitability might be limited for snow applications.

A general disadvantage of scatterometer and passive microwave observations is the spatial resolution. Near coastal regions are influenced by variations in sea ice and sea surface roughness. Masking is required which leads to exclusion of large parts of the near coast land area. This is more pronounced for SMOS than for ASCAT. Several in situ sites of the Yamal transect could be therefore not used for evaluation (Figure 14, #12 and #13). Some mountain regions are also excluded in the SMOS NPR product what may reduce the size of detected ROS as was the case for the Alaska 2021 event (Figure 15). This adds to the issue of data gaps of SMOS due to the polar orbit and swath width at lower latitudes. RFI is permanently high for some regions (see e.g. masked areas inland in Figure 13) and can occur occasionally elsewhere as is demonstrated in the SMOS coverage

**Table 4.** Overview of assessment strategies and results in published studies on ROS using satellite records. AWS - Automatic Weather Station, NARR - North American Regional Reanalysis, ERA - Reanalyses of the European Centre for Medium-Range Weather Forecasts, GPS - Global Positioning System.

| Study | Sensor and Approach | Region and time period | Assessment with meteorological records | Animal populations | Validation |
|---|---|---|---|---|---|
| Grenfell and Putkonen (2008) | SSMI/I, event description | Banks island, one event Oct 2003 | AWS Sachs Harbour, temperature and pressure | muskox herd numbers | visual interpretation (scatter plots) |
| Bartsch et al. (2010a) | QuikScat, snow structure change | Northern Eurasia, 2000-2008, Nov-Feb | 8 AWS, snow and temperature, 1 community documented event on south Yamal 2007 | reindeer herd numbers | visual interpretation (scatter plots) |
| Bartsch et al. (2010c) | QuikScat, snow structure change | circumpolar, 2000-2008, Nov-Feb | | | refers to Bartsch et al. (2010a) for evaluation |
| Semmens et al. (2013) | QuikScat/AMSR-E, snow structure change & wet snow | Yukon basin, 2003-2009, full winter | 2 AWS for visualization; NARR | | on average 30% of events by ROS, others due to fog or higher temperatures only |
| Wilson et al. (2012) | Quikscat, snow structure change | Alaska, 2000-2008, Nov-Feb | 29 AWS (SNOTEL); NARR | | 52% correspondence with NARR regarding single events; AWS 83% correspond to days with T>0°C |
| Forbes et al. (2016) | QuikScat, ASCAT snow structure change | southern Yamal, two events Nov 2007 and 2013 | 2 community documented events ERA-interim | reindeer herd numbers | visual interpretation (maps from reanalysis) |
| Dolant et al. (2016) | AMSR-E, wet snow | 3 sites in northern Quebec, 2010/11 | 29 community observations from three places (near coastal) from one winter | | 72.3% overall accuracy for observations, omission 17%, commission 27% |
| Sokolov et al. (2016) | AMSR-E, wet snow | southern Yamal, one event Nov 2013 | 11 AWS; 1 community documented event | reindeer herd number, predator reproduction number | reindeer mortality 10 times higher, increased numbers of red fox and presence in new areas |
| Langlois et al. (2017) | SSMI/I,AMSR-E, wet snow & snow structure change separately | 18 high Arctic Canadian Islands, 1979-2011 | | caribou herd numbers by island | visual interpretation (scatter plots) |
| Pan et al. (2018) | AMSR-E, wet snow | Alaska, 2003-2016, Nov-Mar | 235 AWS sites (snow, temperature, precipitation, humidity, due point); community observations Fairbanks, 12 events in six years | | AWS overall accuracy 85.9 %, 18% omission; Observers: 75%-100% per winter (1-3 events) |
| **This paper** | ASCAT/SMOS, snow structure change & wet snow | circumpolar, 2011-2019, Nov-Feb | AWS Sodankylä, snow, temperature, LRI; ERA5 for visualisation; 81 snowpit locations in Finland, Norway and Russia | GPS collar information for caribou | March snowpits 76% overall accuracy, omission 21%, commission 12 % |

example in Figure 7 (lower values e.g. over parts of Scandinavia). The impact might be less at lower incidence angles. For larger incidence angles, as used for wet snow observation, the area which affects SMOS observations is larger than for smaller incidence angles. Further investigations are needed to test if the sensitivity of NPR to wet snow is sufficient at small incidence angles in order to fill these gaps.

Additional passive microwave sensor records such as from SSMI/I, AMSR-E and SMAP should be considered to increase the robustness of the wet snow detection. False detection with combination with passive microwave (AMSR-E) at shorter wavelengths has been, however, documented in relation to influence of fog on the snow surface properties (Semmens et al., 2013).

Differences and uncertainties in the location of grid point centres between the wet snow and snow structure change products may lead to lower precision in addition to the grid size issue. Where available, SAR could be used to complement the wet snow retrieval and even provide more detailed information on the actual boundaries of events. C-band SAR could in addition be used to identify snow structure change where temporal sampling is insufficient for reliable wet snow retrieval. Both commonly available polarization options from Sentinel-1 (C-band HH+HV, Figure 14, and VV+VH) show similar $\Delta\sigma^0$ behaviour like ASCAT. The combination of the different polarization types needs to be investigated since SAR data acquired across the Arctic are inconsistent (Bartsch et al., 2021).

In some cases high SMOS NPR without change in ASCAT backscatter can be observed. This occurs for example during November 2012 when further multiple warm phases occurred after the first ROS which lead to a pronounced backscatter increase (Figure 3). ASCAT backscatter varies strongly during soil freeze-up before snow is on the ground. This may lead to false event detection. Also SMOS NPR can increase above the defined threshold during this period. The approach should be therefore only applied for full snow cover. We limited the full Arctic comparison (Figures 7 and 9) to November to February to ensure snow presence. This, however, excludes events such as on Banks Island (Putkonen and Roe, 2003) or the first Yamal event in 2020 (Figure 4).

A high LRI without strong change in ASCAT backscatter or SMOS NPR can be also documented for the Sodankylä site. Records are more noisy for both sensors during such periods, but thresholds were not exceeded. For SMOS, this may relate to acquisition timing. Saturation due to the preceding event (resulting in about -10 dB for $\sigma^0$) might be a potential reason in case of ASCAT (Figure 3). Events are also not reflected anymore if a certain level (degree of snow metamorphoses) is reached. This is mostly an issue for regions with continuous ice cover which have been excluded from this analyses. Ice sheets show a rather uniform backscatter level at frozen conditions and ROS can be detected through surface melt observations only. Freund and Bartsch (2020) found three major melt events on the Greenland ice sheet in midwinter based on Seawinds Quikscat (26.11.-30.11.2005, 17.11.-21.11.2007 and 01.-04.11.2008). Ice sheets should be included for full Arctic land characterization but only wet snow retrieval applied. Similar issues may apply for sea ice.

The temporal window size (plus minus three days) applied for combination with SMOS may result in false ROS identification when the temperature drops strongly right after a wet snow detection. This has been for example the case at the end of November 2016 at Sondankylä (Figure 3). The window based approach is, however, needed in order to account for low precision in

capturing the date of snow structure change with ASCAT. The detection difference in case of the eight documented ROS for Fairbanks (Pan et al., 2018) within the analyses period ranges by $\pm 2$, with on average 0.6 days indicating early detection.

Metop ASCAT is available since 2007, but the tested masking scheme is limited to 2011 onward due to later launch of SMOS. A combination with wet snow products from e.g. SSMI/I or AMSR-E could be used to fill the gap. Passive microwave observations might be also of benefit for ice layer detection (Langlois et al., 2017) but validation in the context of ROS is still lacking.

In the case of known events, ASCAT results alone could be used as demonstrated for the Alaska event (Figure 15). Care needs to be taken for regional comparisons and climate record data retrieval. An increase of events for the entire Arctic can not be observed with the used records (Figure 9). Regional extremes and recent increase of events can be however identified as in the example of Alaska, specifically on the Seward peninsula (Figure 16). The average events per grid point in 2021 have been clearly above the preceding December values during the ASCAT/SMOS record period.

## 6.4   Impacts on animal populations

Western Alaska and Western Siberia have been discussed before regarding potential impact on the environment including hydrology and wildlife (Bartsch et al., 2010c; Semmens et al., 2013; Forbes et al., 2016; Sokolov et al., 2016). The two events described for Yamal during the ASCAT period have been the only major events. ROS in this region are rare but have a severe impact especially as they occurred in the first part of the winter which is an issue for reindeer, for example, in the region. All events occurred over the period of several days, progressing spatially. The pattern was different between 2013 and 2020. The direction was West to East in 2013 and partially North to South in 2020. The overall extent and shape (W-E band from Franz-Joseph-Land to the Tazovskiy Peninsula, Figure 13) of the 2013 event agrees with regional AMSR-E2 retrievals by Sokolov et al. (2016). Forbes et al. (2016) suggested a link to sea ice conditions in November for the trigger of such events. Sea ice concentration was lower than average in the proximity of Yamal in both years (Figure 13 and Figure 4), but specifically low towards the west and north in 2020 which might explain the spatial extent and progression of the ROS.

From 2010-2017, caribou regularly migrated south on their fall migration to the Seward Peninsula, but not since. The distribution of caribou observed during the winter of 2021/2022 was different than 2010-2017 (Figure 15). They stayed further northeast (including the Kobuk River plain) and did not disperse to the Seward Peninsula. The Seward Peninsula was among the most affected regions during the two Alaska events in December 2021. If, in January 2022, caribou migrated to where they were found in January 2010-2017, they would have experienced 40.5 % greater $\Delta\sigma^0$ than the caribou actually experienced in January 2022.

Sea ice extent was also lower than average in the case of the Alaska 2021 events (Figure 15), specifically for the second phase. Mean event size for Alaska has been previously quantified as 469 km$^2$ and at maximum approximately 363,000 km$^2$ for January based on the QuikScat record 2001–2008 (Wilson et al., 2012). December maximum reached 220,000 km$^2$ which corresponds approximately to the events observed in December 2021. The situation was unique throughout the ASCAT/SMOS record period (2011-2021) for the Seward Peninsula (Figure 16) and may have contributed to the observed distribution of caribou in the region in early 2022 (Figure 15).

## 7 Conclusions

The utility of standardized snowpit records, specifically information on ice crusts, has been demonstrated for the first time for evaluation of satellite derived rain-on-snow events. The presence of crusts (pencil/knife test) coincides with events detected with the ASCAT/SMOS fusion approach throughout the winter season. In cases without events, no crusts or only thin ones are observed. Below average regional sea ice concentration at the timing of events can be observed for all examples (Yamal as well as Alaska events), but the number of studied events is too low to infer any linkage.

C-band scatterometer based retrieval at VV polarization has been shown feasible, but it requires post-processing to reduce false detection in case of dropping temperature. The combination of snow structure change and wet snow information provides a solution in this case. The potential and limitations of the use of L-band passive microwave observations has been demonstrated. The combination with SMOS is, however, recommended only for application north of 65°N due to coverage issues. Other passive microwave sensors with better spatio-temporal coverage may allow extension of the results. The magnitude of specific known extreme ROS events can be, however, documented by use of ASCAT alone, what allows the inclusion of regions south of 65°N and years before SMOS availability. Such results can be used for documentation of impacts, as we demonstrated for caribou migration patterns in Alaska.

Initial results of the SAR analyses indicate the potential of C-band SAR at all polarization combinations. Both HH/HV (Yamal) and VV/VH (Finland) SAR records show similar behaviour like ASCAT VV. This is crucial since SAR data are not acquired with consistent polarization across the Arctic. A fusion of scatterometer and SAR records for detection of the snow structure change might allow for a more detailed impact assessment than with ASCAT alone.

Discrepancies between C-band and Ku-band radar in boreal regions for detection of snow structure change can be observed which may relate to issues with both data types. Frequency and polarization may play a role, which should be considered in future studies. Nevertheless, similarity regarding number of events and spatial patterns between different products (QuikScat, Oceansat-2 and ASCAT/SMOS) can be observed for tundra regions which allows analyses over more than 20 years. Trends for the Arctic cannot be derived but extremes identified and documented.

*Data availability.* Step 1 (ASCAT only) and step 2 (combination with SMOS) results will be published in seasonally aggregated format and daily maps in due course on PANGAEA.

*Author contributions.* AB has developed the concept for the study, analysed all results and wrote the first draft of the manuscript. HB contributed to satellite data processing and writing of the manuscript. GP and XM contributed to the satellite data processing. KR preprocessed SMOS data and contributed to the manuscript. LL, AS, and PO collected in situ snow data and contributed to manuscript writing. DE and ES provided expertise on snowpit observations and contributed to the writing of the manuscript. KCJ processed collar records and contributed to the writing of the manuscript.

*Competing interests.* The authors declare no competing interests

*Acknowledgements.* The project has received funding under the European Union's Horizon 2020 Research and Innovation Programme under Grant Agreement No. 869471 (CHARTER). This work was further supported by the European Space Agency CCI+ Permafrost project.

Oceansat-2 data is obtained from the NASA sponsored Scatterometer Climate Record Pathfinder at Brigham Young University courtesy of David G. Long.

Yamal-campaign data was collected during an expedition organized by the Yamal Government. We thank Aleksandr Volkovitskiy for help 595 in sampling along snowpit transect during the Yamal-campaign.

Rolf A. Ims, Nigel Yoccoz, Hanna Böhner and Jan Erik Knutsen from UiT respectively initiated the collection of snow profiles (RAI and NY), cleaned the data (NY and HB), provided logistics help (JEK), and participated the fieldwork (RAI, NY, and JEK).

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

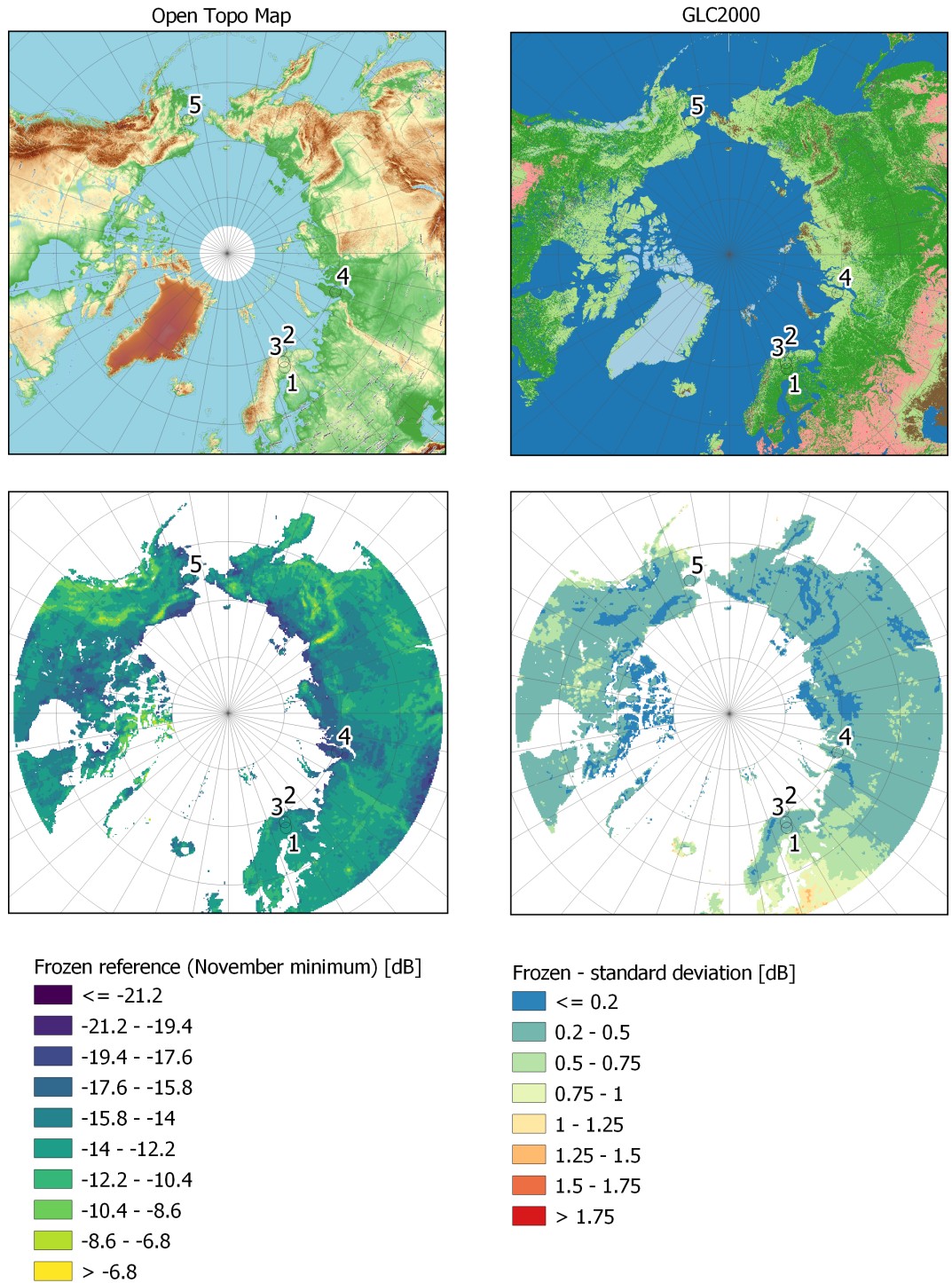

**Figure 2.** Characteristics of ASCAT analyses extent. Upper left: digital terrain model (source: ©OpenStreetMap-Contributors, SRTM | Mapview: ©OpenTopoMap (CC-BY-SA)), upper right: Landcover (source: Global Land Cover 2000 database. European Commission, Joint Research Centre, 2003; dark green - forest, bright green - shrubs to sparse vegetation, brown - bare, grey - burned forest, coral - cultivated, red - artificial). Metop ASCAT characteristics considered for ROS retrieval: Frozen soil reference (lower left) and standard deviation for November to February (lower right) for $\sigma^0$. Circles: in situ and rain on snow case study sites: 1 - Sodankylä, 2 - Varanger peninsula/Upper Komagdalen, 3 - Saariselkä/Niilanpää, 4 - Yamal peninsula, 5 - Seward peninsula.

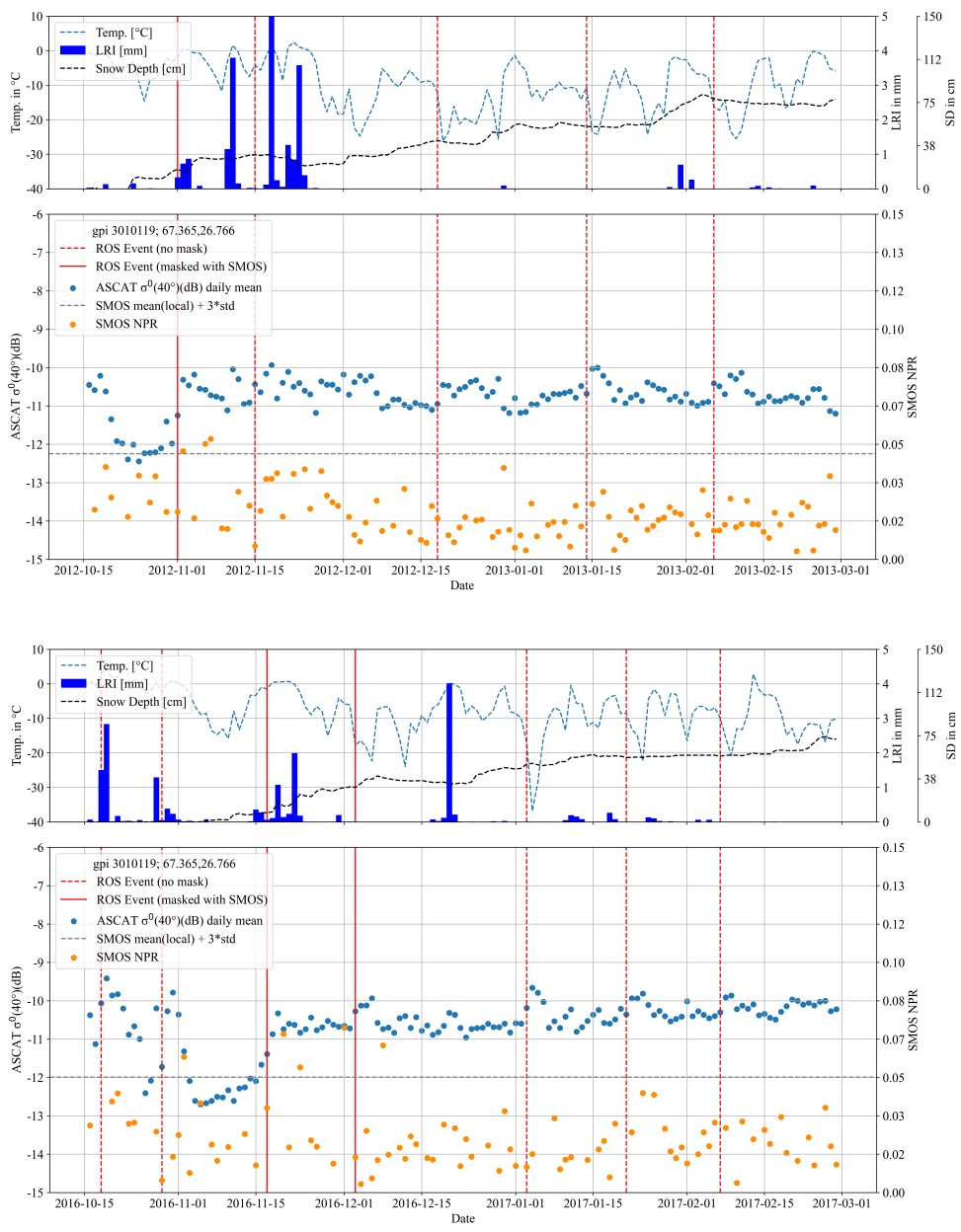

**Figure 3.** Examples of the influence of temperature drops on the rain-on-snow detection based on ASCAT for Sodankylä. Liquid precipitation (LRI) is shown together with air temperature and snow depth (SD) from the Automatic Weather Station for two selected winter periods. SMOS NPR and the masking threshold (derived from local mean NPR and the high latitude standard deviation, see Table 2) is shown in addition as grey horizontal dashed line. Vertical lines represent potential rain-on-snow events based on ASCAT only, with solid lines for periods confirmed with high SMOS NPR (wet snow).

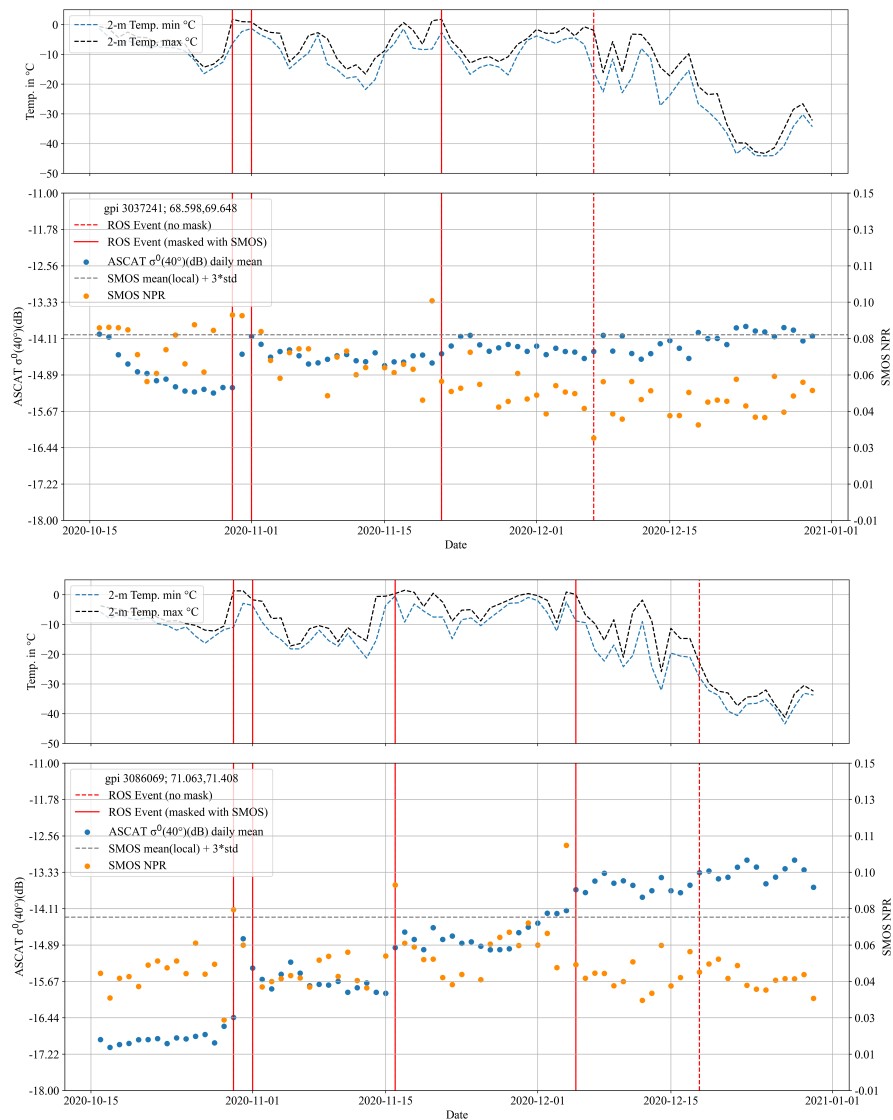

**Figure 4.** Time series examples for the event series in early winter 2020 on the Yamal peninsula, a) south, #3 , b) north #18. For location see Figure 5.

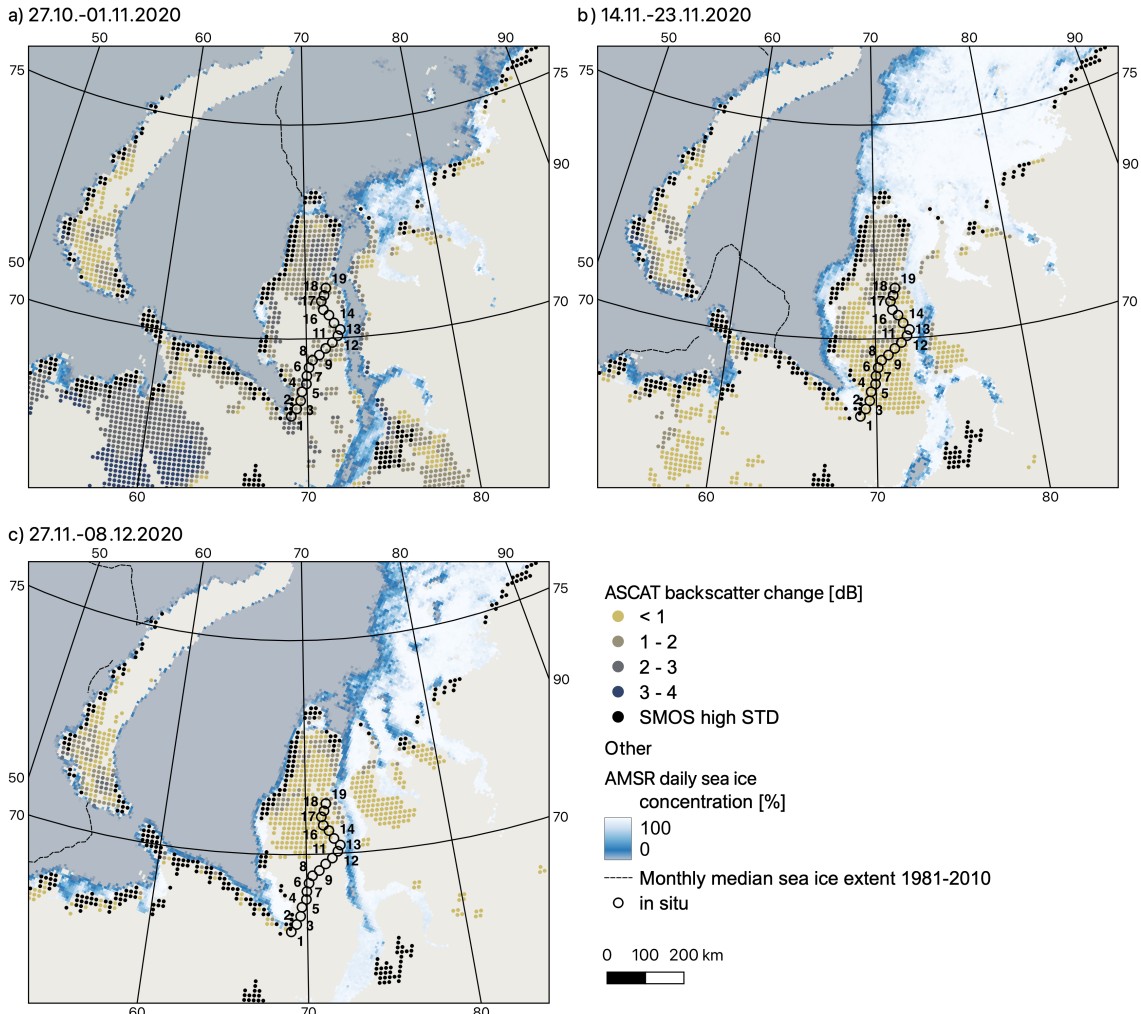

**Figure 5.** Cumulative ASCAT $\Delta\sigma^0$ (in dB, incl. SMOS masking for wet snow and RFI) and AMSR sea ice extent (source: University of Bremen, Spreen et al. (2008)) for event series in early winter 2020 on the Yamal peninsula. a) 27.10.-01.11.2020, sea ice extent 26.10.2020 and October monthly mean, b) 14.11.-23.11.2020, sea ice extent 13.11.2020 and November monthly mean, c) 27.11.-8.12.2020., sea ice extent 27.11.2020 and December monthly mean. Location of in situ points for snow surveys in February 2021 (Figure 14).

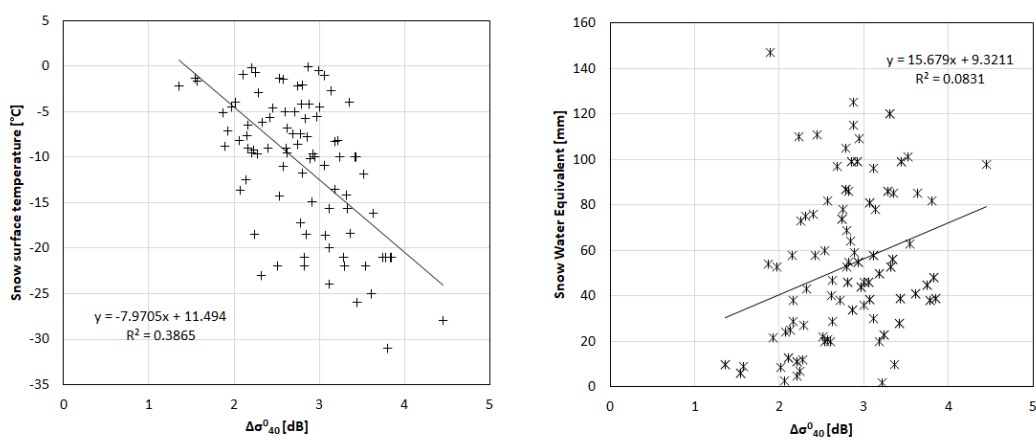

**Figure 6.** Comparison of Metop ASCAT C-VV $\Delta\sigma^0$ with (a) snow surface temperature and (b) snow water equivalent. Source: snowpit data at Sodankylä BOG site 2009-2015, temperatures below 0°C and SWE > 0.

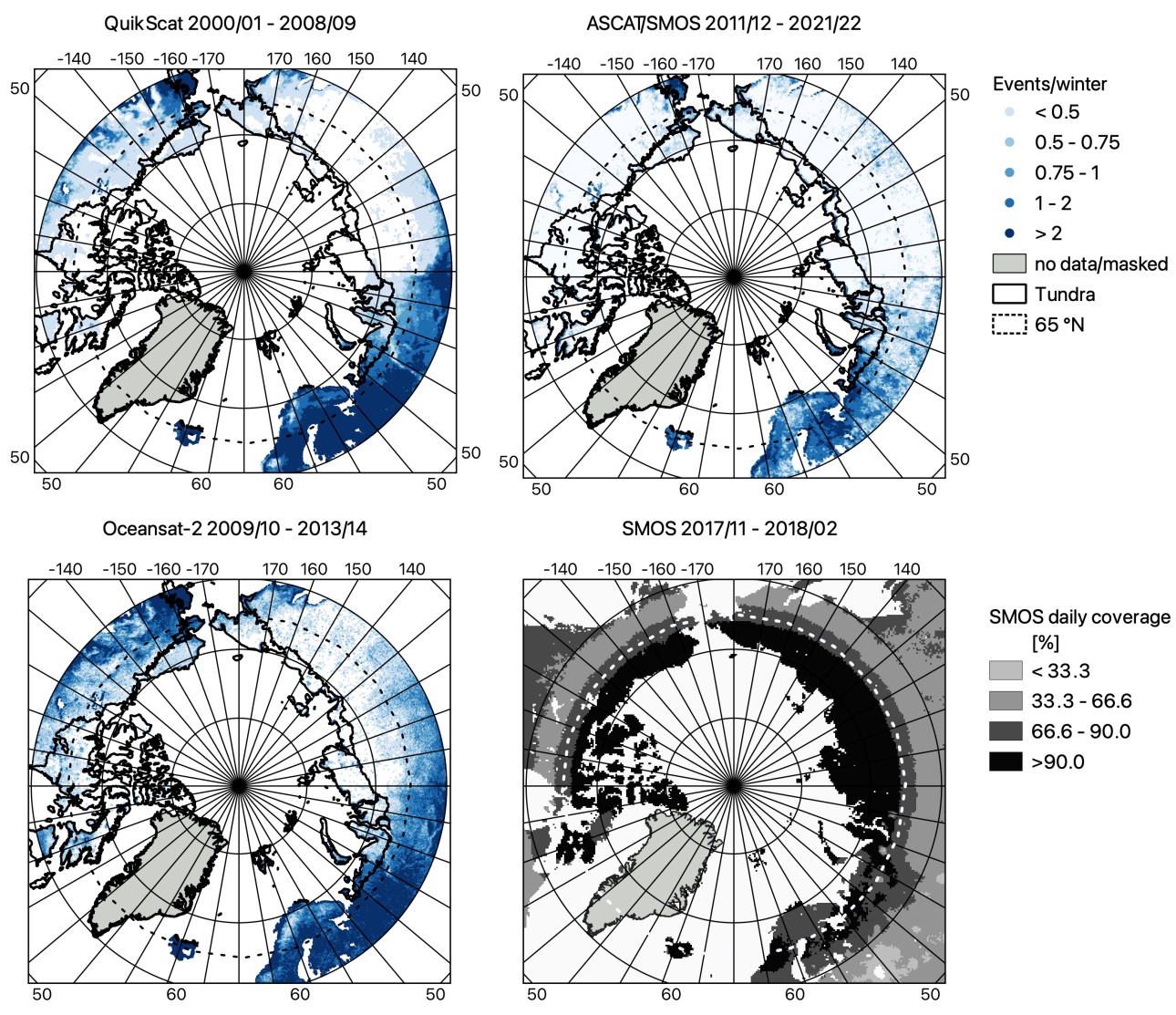

**Figure 7.** Average potential events per winter and grid cell for each product and example for SMOS data coverage. Boundaries for time series analyses extent (Figure 9) are indicated (tundra extend source: CAVM, Raynolds et al. (2019)).

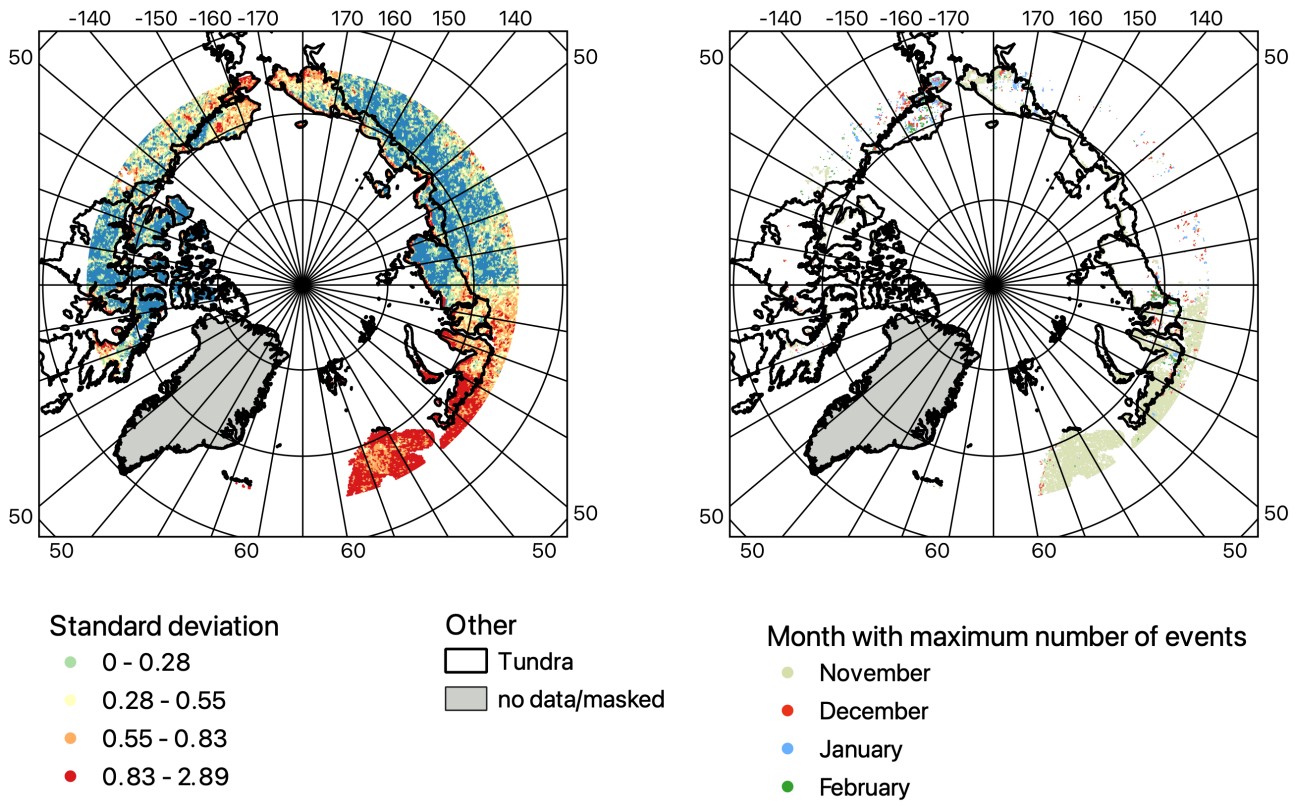

**Figure 8.** ASCAT/SMOS derived potential events per winter (November to February, north of 65°N) and grid cell: left - standard deviation, right - month with maximum number of events (tundra extend source: CAVM, Raynolds et al. (2019)),

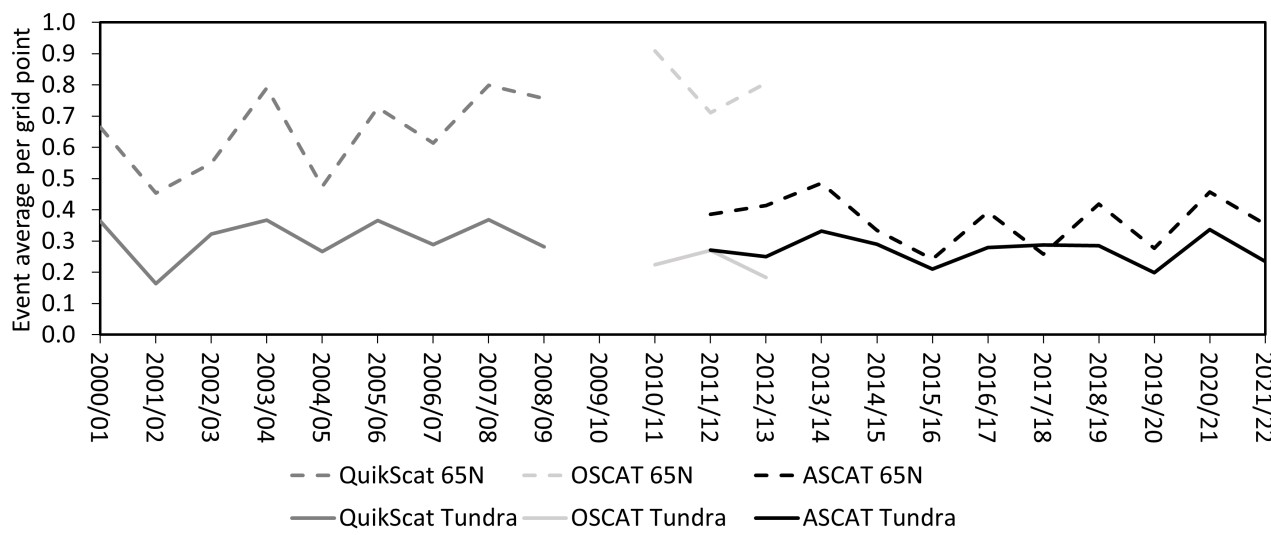

**Figure 9.** Average potential high latitude ROS events per winter and grid cell (resampled to QuikScat 10 x 10 km grid) for overlap area of are north of 65°N (excludes Greenland and masked areas in the ASCAT/SMOS results) as well as for tundra only (extend source: CAVM, Raynolds et al. (2019)).

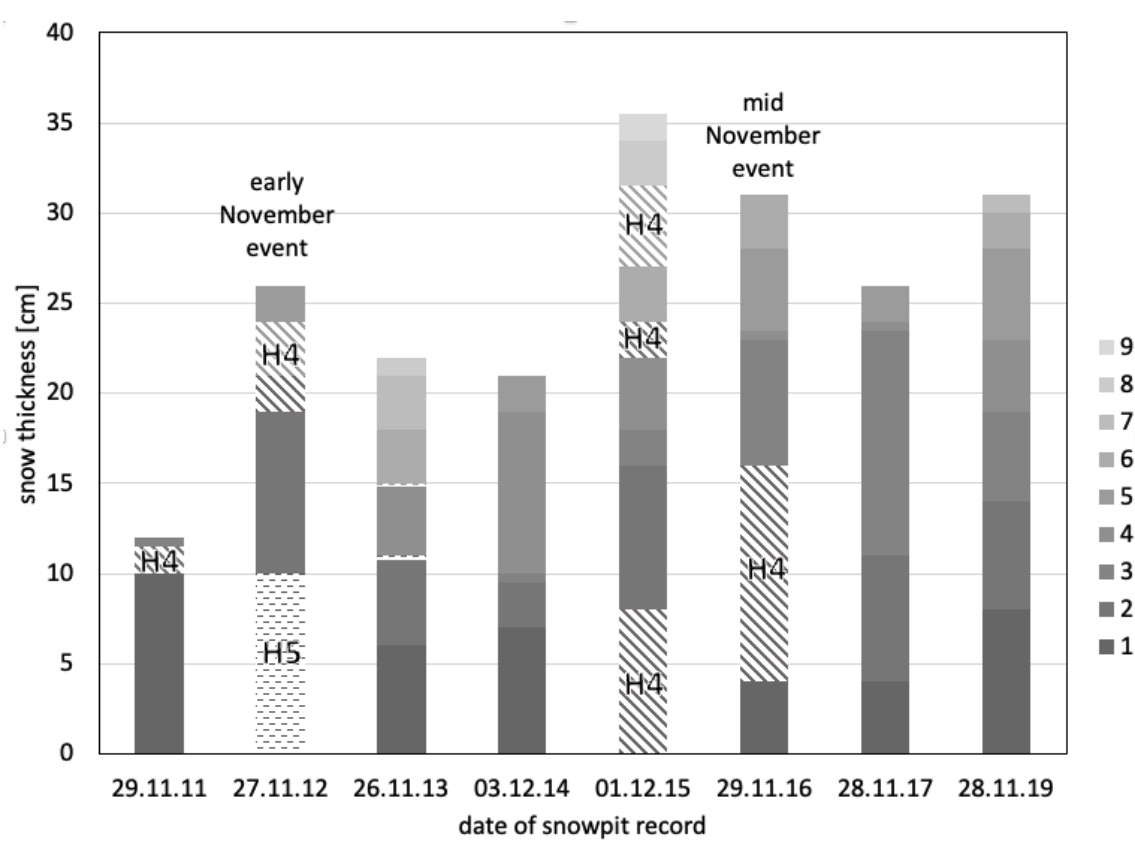

**Figure 10.** Snowpit observations at the IOA site (Sodankylä) in late November/early December. Years with rain-on-snow events detected through ASCAT as well as SMOS are indicated (see also Figure 3). Individual layers are represented through grey gradients. Hardness 4 layers are highlighted with diagonal hashes and hardness 5 with horizontal dashed lines.

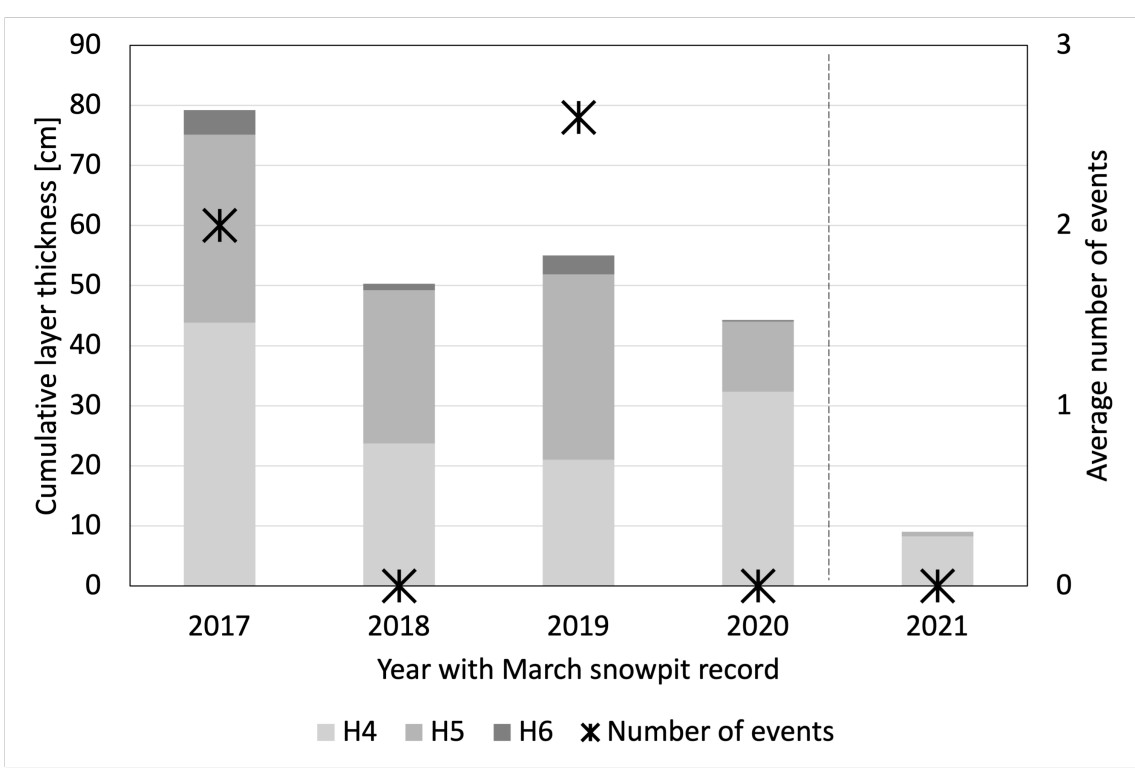

**Figure 11.** Average cumulative snow layer thickness for hardness classes 4-6 (for definition see Table 1) in snowpits on Varanger (66 locations in six ASCAT grid points; mid to end of March 2017-2020) and at Saariselkä (two locations in one ASCAT grid point; 18.03.2021). Secondary axis: number of potential rain-on-snow events between November and February (average from six grid points for 2017-2020). Snow depth range: 20-234cm on Varanger and 42-66cm at Niilanpää.

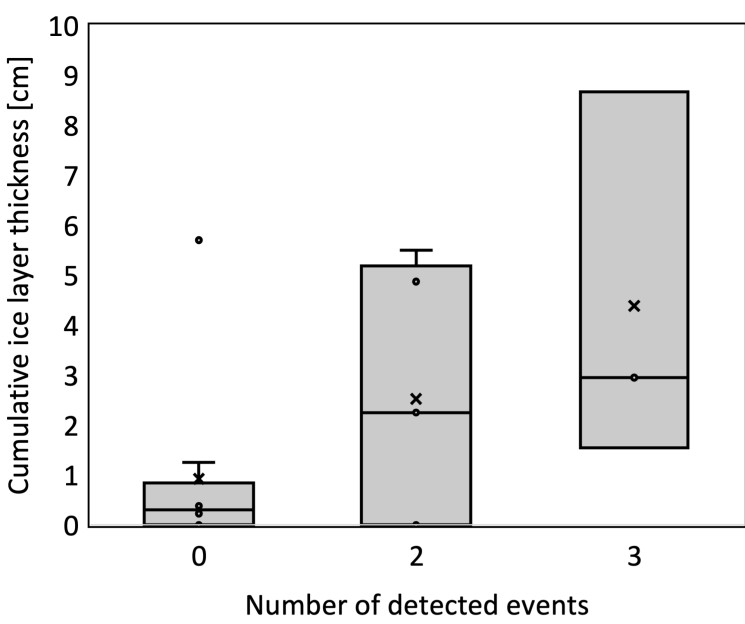

**Figure 12.** Boxplots for ASCAT grid point average cumulative snow layer thickness for hardness class 6 (for definition see table 1) from snowpits at Varanger (66 locations in six ASCAT GPIs; mid to end of March 2017-2020) and at Saariselkä (2 locations in one ASCAT GPI; 18.03.2021). Separation for number of detected potential rain-on-snow events between November and February and sum for class 6. Snow depth range: 20-234cm on Varanger and 42-66cm at Niilanpää.

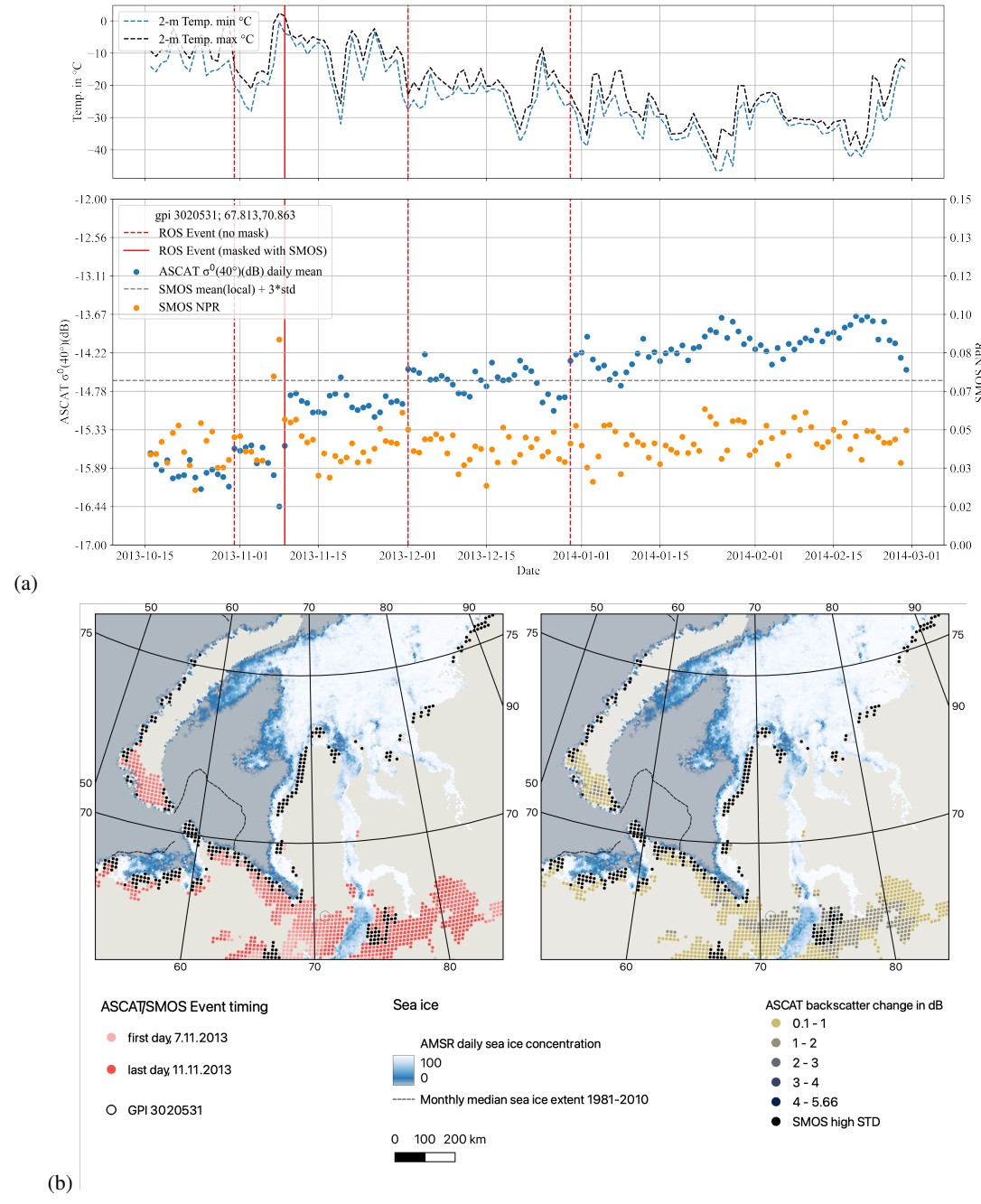

**Figure 13.** Example for a grid point on southern Yamal documenting the 2013 event (as described in Forbes et al. (2016)). (a) Time series of ASCAT backscatter and SMOS NPR. Air temperature data have been obtained from reanalyses data (ERA5). Only the detected event around 10th of November has been caused by a ROS (with a high SMOS NPR at the same time). All other events correspond to temperature drops. For location see black circle in (b). b) Grid points with an event in November 2013 and sea ice concentration (9.11.2013 and long-term November average, source: University of Bremen, Spreen et al. (2008)). Left: event date by grid point, right: ASCAT $\Delta\sigma^0$.

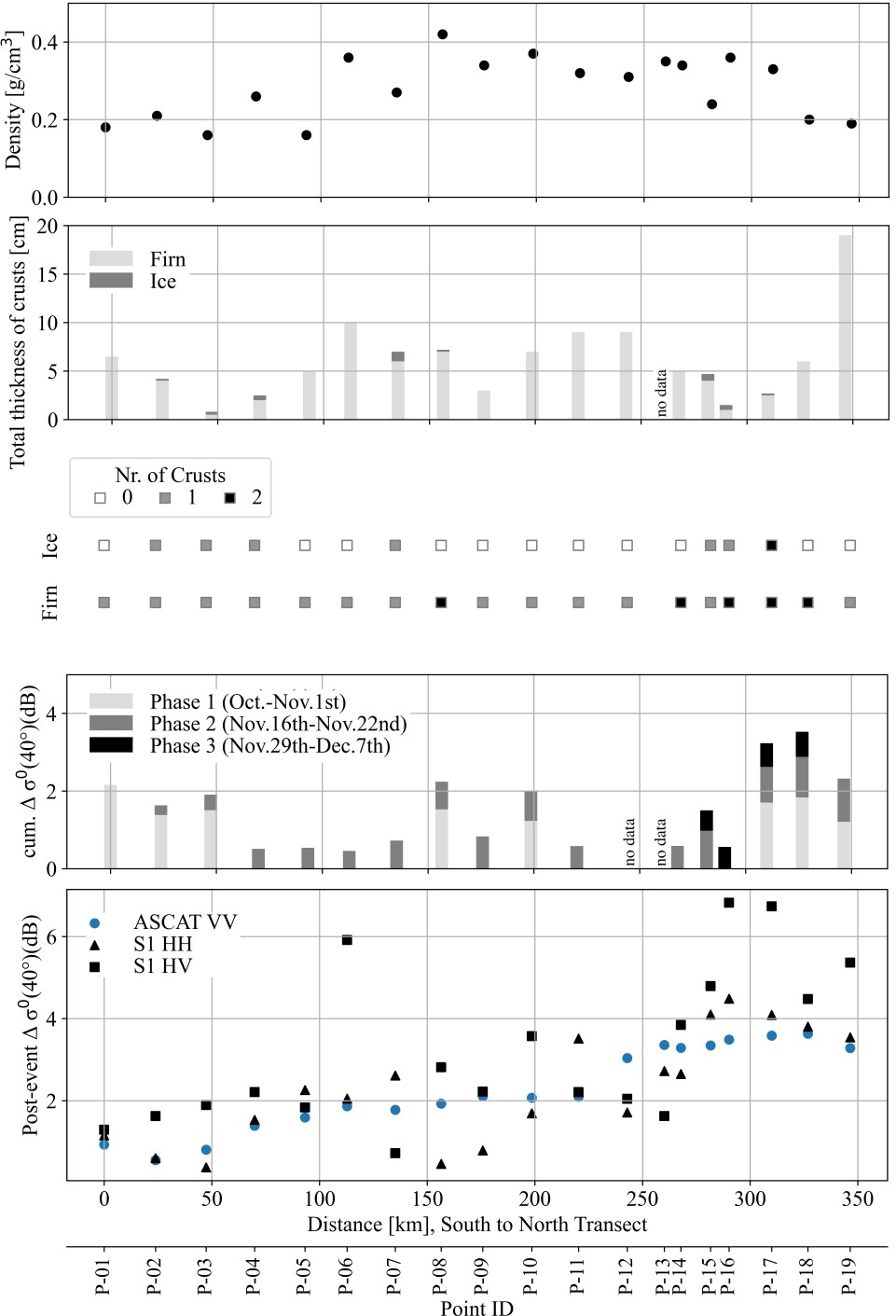

**Figure 14.** Documentation of snow conditions after the ROS events across Yamal end of 2020 including in situ measurements obtained in February 2021, ASCAT $\Delta\sigma^0$ along the in situ transect by event (difference before and after) and $\Delta\sigma^0$ from ASCAT and Sentinel-1 during the week after the last event (increase since November). For location of points see Figure 5. No event data is available for coastal points 12 and 13 due to SMOS data gap.

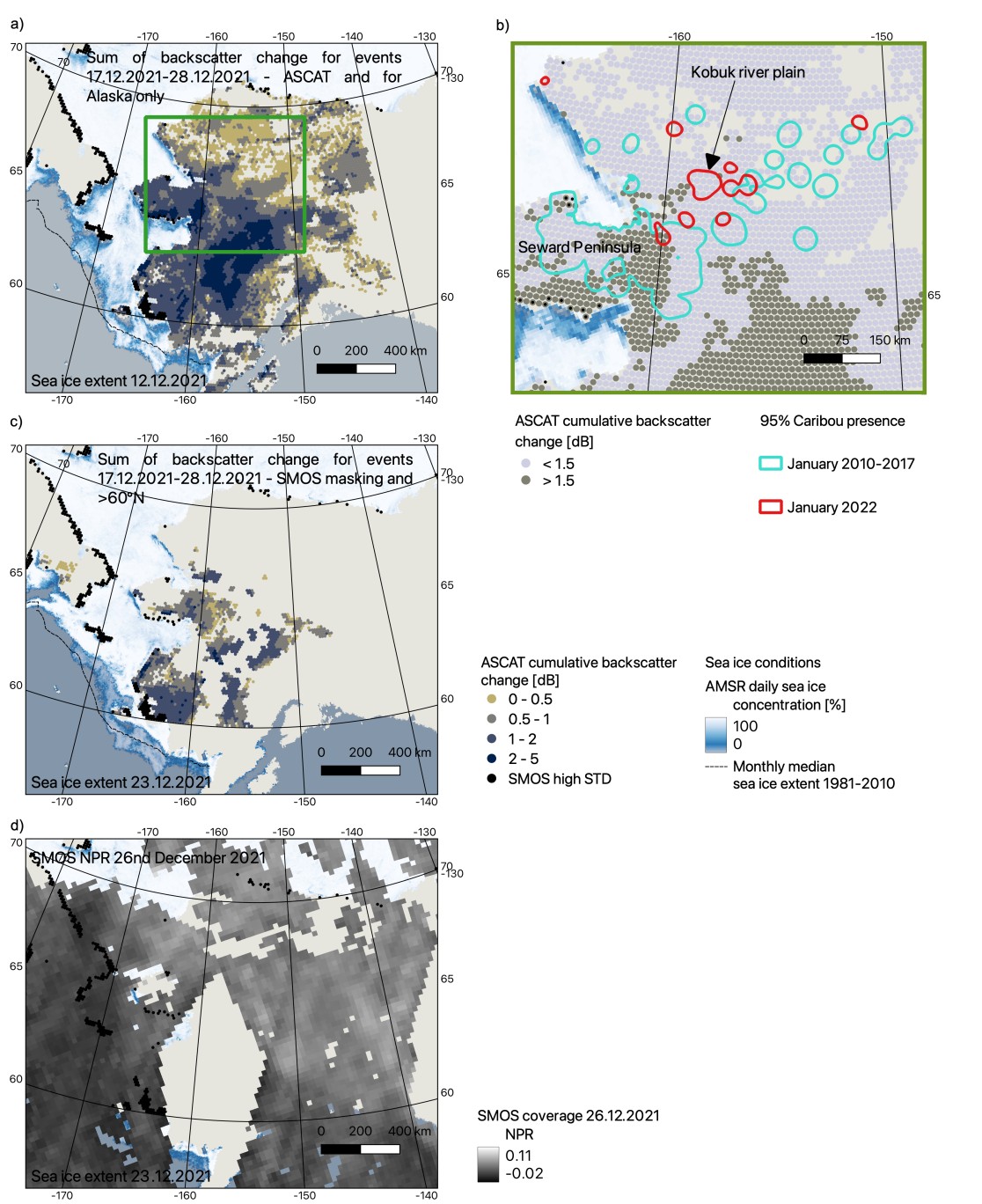

**Figure 15.** Rain on snow event in Alaska December 2021: a) cumulative ASCAT $\Delta\sigma^0$ for both events before masking with SMOS and sea ice extent before the first event; green rectangle - extent of map (c); b) after masking and sea ice extent at the beginning of the second event, c) separation of cumulative ASCAT $\Delta\sigma^0$ > 1dB and areas of caribou presence January 2010-2017 versus 2022; d) example for SMOS NPR availability from a single day. Source of sea ice data: University of Bremen, Spreen et al. (2008).

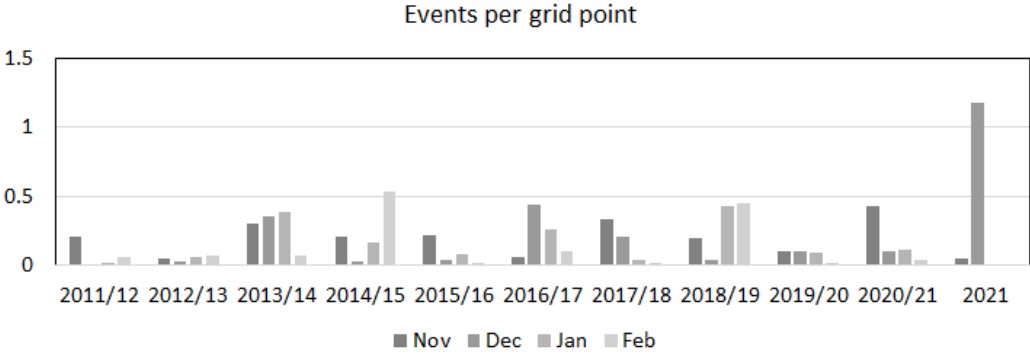

**Figure 16.** Example for monthly aggregated events. Number of events per grid point for the Seward Peninsula (for location see Figure 15). SMOS masking (wet snow and RFI) has been applied.