# Peer review of "Towards long-term records of rain-on-snow events across the Arctic from satellite data"

_EGUsphere, 2022_

## Referee Comment (RC1)

Warming temperature showed already sign of increased extreme events such as Rain on snow (ROS) in the Arctic. ROS event occurs from a rain event usually with temperature around the melting point and results in ice crust after temperatures goes down and the liquid water on the surface refreezes. The study presents a way of detecting such events by using a novel method from combined satellite C-band radar and L-band passive observations to detect liquid water. They evaluate the retrieval of several events in the Arctic. The usage of high-resolution SAR imagery from Sentinel 1 is particularly interesting and novel since such events can occur locally. However, I believe this added benefit could be more highlighted. I also think this work lacks a quantitative validation of the ROS retrieval from ice layer I snowpits, maybe % of commission/omission from the algorithm could be calculated. I would also propose to compare to another passive ROS detection algorithm to evaluate the benefit of the method. The combined method of C and L band is interesting. I think the focus of the paper could be narrowed, it is a bit broad and makes the paper harder to understand. I would aim to present the method, then evaluate and not try to make a broad-scale statement on ROS event since it is not properly evaluated yet. Overall, I think this article should be published once modifications are made.

Specific comments:

L35. I would not use the term "*aging of snow*" since it is not accurate. Please refer to the vapour flux from temperature gradient.

L38. "The mapping of snow changes afterwards instead of wet snow circumvents", I'm not sure what is meant in the sentence. Please consider modification for improved clarity.

L40. Include citation on wavelength and snow grain size.

l65. Do you mean "*With ROS, associated*"

L77-80. Perhaps the objective should be modify or addressed more clearly in the conclusion. Were you able to correctly answer (1) with this method? How was (3) evaluated?

L150. Consider adding a statement on how these data can be subjective and what was done to avoid this.

L207. Please clarify this sentence "ROS using wet snow from C-band". Do you mean wet snow detection?

L.246. Please reword the beginning of the sentence.

L248. Algorithms for ROS detection using 37 and 19 GHz are also sensitive to dry snow surface change into ice crust and ice layer. Consider using does to improve the algorithm since L band is useless when no liquid water is present.

L306. Revise figure reference.

L310. Consider comparing your method to a passive-based ROS (Dolant et la., 2016, Pan et al., 2018) retrieval to show the improvement your method could deliver.

L324. Revise table reference.

L521-523 Can you provide a quantitative validation of the method to detect ROS events?

L527. the phrasing with the comma is confusing, do you mean ... "*play a role on what should be considered*"

L526. Maybe consider using a passive observation with 19 and 37 GHz to improve sensitivity to ice crust and dry snow surface change. Once the liquid water is frozen and the temporal timing of the ROS event could not be detected with SMOS, those frequencies could help to detect surface change while C-band can provide info at high resolution.

L529. "*The magnitude of specific extreme events can be documented by the use of ASCAT alone, without fusion with SMOS.*" I thought you showed you need wet snow detection and ASCAT alone cannot detect ROS.

Figure 6. This figure is hard to understand. what are H4 1, H4 2 and H4 3? Why not add all layers so we have a better understanding of the whole snowpack?

Figure. Overall, the labels are hard to read since the font is so small. Consider increasing font.

Dolant, C., Langlois, A., Montpetit, B., Brucker, L., Roy, A., & Roy, A. (2016) Development of a rain-on-snow detection algorithm using passive microwave radiometry. Hydrological Processes, 30, 3184-3196. DOI: 10.1002/hyp.1082

Pan, C. G, Kirchner, P. B, Kimball, J. S, Kim, Y. & Du, J. (2018). Environ. Res. Lett. 13 075004, DOI : 10.1088/1748-9326/aac9d3

---

## Referee Comment (RC2)

In this manuscript "Towards long-term records of rain-on-snow events across the Arctic from satellite data" the authors have presented an approach to mapping ROS events by combining observations from C-band scatterometer, which are indicative of snow structure change and L-band passive microwave radar, which is sensitive to the presence of wet snow. The results highlight the added value of using L-band observations to filter out false detections from the ASCAT C-band data which can occur as a result of temperature drops. In addition the authors have made use of additional data sources to support the remote sensing detections, including snow pit data, AWS measurements and caribou data. The use of C-band Sentinel-1 SAR observations was also described and presented, but in my opinion these data played a much smaller role in the data analysis than what I was expecting from earlier on in the manuscript. Overall the method is convincing and the datasets produced are valuable in providing a better understanding of ROS occurrence across the Arctic. However, I have a few general and specific comments which should be straightforward to address before the manuscript is published.

1. Area of study: it is apparent from the figures that the authors have applied the approach to not only the entire Arctic but also to land areas extending much further south (eg., Figure 5). However, it is stated in the conclusion that the approach is only recommended for regions north of 66 deg due to coverage issues with SMOS. Perhaps the authors should revise the boundaries of the areas for which ROS detections are presented, or comment on how representative the ROS data are for the lower latitude areas shown in Fig.5?
2. The authors have described and presented a wide range of different types of observations, but I think that some of the datasets used do not really add much to the overall goals and conclusions of the study. While I see the need for observations to validate/support the remote sensing data, I think the use of too many different observations, each with their own considerations for ROS detection, makes it at times difficult to follow the main objectives of the study. I would for example recommend reconsidering whether the use of the caribou data are really necessary. While I think the presentation of Sentinel-1 observations is interesting, I don't think it featured enough in the results to be worth including in the data descriptions/method. Perhaps the authors might consider a follow-up study which focuses primarily on Sentinel-1 instead of including it here?

Specific comments

Line 175: why were different terms/hardness scales used at Yamal compared to the Scandinavian sites? Why not just use a standard scale for all sites?

Line 296: "location specific threshold" - does this mean that a threshold is determined for individual pixels, or for regions?

Table 2: Events represent November 2021 to February 2022; why are values from only 1 year/winter of observations used?

Figure 1: missing?

Figure 2: Please consider splitting into 2 figures as it is very difficult to see the circles showing the reference sites

Figure 4: Could the authors comment on the event confirmed by SMOS occurring in the start of December 2016? Here the AWS data show very low temperature (approx. -15 deg.C), no

precipitation and increasing snow depth in the following days. What could be the reason for detection of wet snow?

Figure 5: Consider splitting into 2 figures; the pan-Arctic maps showing ROS frequency are quite small but show very interesting data. Also consider increasing the symbol size in the legend.

Figure 8: Legend symbols are far too small. Also move the legend for ASCAT backscatter change closer to the actual figure showing backscatter change (right panel of Fig. 8b)

Figure 10: Increase the symbol and text sizes in the legend

Figure 11: Missing labels (a) and (b)

Figure 12: Consider removing the caribou observations (panel c) and increase legend/symbol sizes elsewhere

Figure 13: I didn't find that this figure showed any useful information, consider removing it

Technical corrections

Line 38: change "afterwards" to "following a ROS event"

Line 42: change "ROS events" to "ROS event"

Line 99: Change "production if" to "production of"

Line 129: Define EASE2

Line 130: Define RFI

Line 161: Change "University Tromso" to "University of Tromsø"

Line 162: "information is collected" -> "information has been collected"

Line 163: "is used since" -> "has been used since"

Line 170: "First snow pits" -> "The first snow pits"

Line 181: recommend changing "instrumented" to "equipped"

Line 199: "what is reflected to a certain magnitude depending on" -> "which is reflected to a certain degree by"

Line 202: "is increasing in" -> "increases"

Line 204: "is significantly decreasing" -> "decreases significantly"

Line 208: "investigated for Norway" -> "investigated for the Svalbard archipelago"

Line 209: "alter" -> "alters" and "even ice layers are forming what allows" -> "the formation of ice layers allows"

Line 226: "found independent" -> "found to be independent"

Line 232: "real part" -> "the real part"

Line 235: "The attenuation of dry snow layer" -> "The attenuation by dry snow layers"

Line 247: "Band rations" -> "Band ratios"

Line 257: "towards south" -> "towards the south"

Line 300: Figure 4 should be referred to after Figure 3

Line 306: something missing: "?"

Line 307: "the finally determined date" -> "the final date determined"

Line 309: Figure 11 is referred to too early? Figure 5 should come after Figure 4

Line 324: Table ?? - missing number

Line 331: outlined in ? - missing a section number

Line 398: Missing word after "in the following" - winter/summer?

Line 435: "what leads to" -> "which leads to"

Line 439: "In same regions" -> "In the same regions"

Line 440: "Problematic are also" -> "Also problematic are"

Line 441: "change is decreasing" -> "change decreases"

Line 449: "what underlines" -> "which underlines"

Line 464: "Near coastal" -> "Coastal"

Line 481: "Both from Senintel-1" -> "Both Sentinel-1"

Line 482: "as ASCAT" -> "to ASCAT"

Line 483: "In cases" -> "In some cases"

Line 500: "In case of" -> "In the case of"

Line 507: "what is an issue" -> "which is an issue"

Line 513: "what might explain to" -> "which might explain"

Line 514: "in case of" -> "in the case of"

Line 516: "what corresponds" -> "which corresponds"

Line 527: "what should be" -> "which should be"

Line 533: "what allows" -> "which allows"

---

## Author Response (AR1)

**Reviewer 1**

R1: Warming temperature showed already sign of increased extreme events such as Rain on snow (ROS) in the Arctic. ROS event occurs from a rain event usually with temperature around the melting point and results in ice crust after temperatures goes down and the liquid water on the surface refreezes. The study presents a way of detecting such events by using a novel method from combined satellite C-band radar and L-band passive observations to detect liquid water. They evaluate the retrieval of several events in the Arctic. The usage of high-resolution SAR imagery from Sentinel 1 is particularly interesting and novel since such events can occur locally. However, I believe this added benefit could be more highlighted.

> Reply: we have now added text on the SAR results to the conclusions and abstract

RC1: I also think this work lacks a quantitative validation of the ROS retrieval from ice layer I snowpits, maybe % of commission/omission from the algorithm could be calculated.

> Reply: It is unfortunately not possible to infer from the snowpit records what the reason for formation of hard layers was. It is very common that wind compaction leads to higher hardness at the sites with the snow pits. This especially applies to snowpits from Varanger which are made towards the end of the winter.
> We have now, however, used data from 66 instead of 4 snow pit location on Varanger and compared them to number of events each winter. In general thickness of H4-6 crusts is higher for higher number of events. We included text on snow compaction for the interpretation of such records in the discussion.
> In case of Sodankylä, where events usually occur in November, we can use snow pits from shortly after the events. The number of events that can be analysed with such a strategy is too small (although in a similar order or higher as used for validation in previous publications on ROS). We have now included an overview on evaluation strategies in previous papers in the discussion in form of a table (Table 4).

RC1: I would also propose to compare to another passive ROS detection algorithm to evaluate the benefit of the method. ... L310. Consider comparing your method to a passive-based ROS (Dolant et la., 2016, Pan et al., 2018) retrieval to show the improvement your method could deliver.

> Reply: Other algorithms for ROS detection from passive microwave L-band have not been published yet to our knowledge. ROS retrieval results from passive microwave records in general have (to our knowledge) so far only documented with figures in publications and derived datasets of events not published. One event was however documented by one of our co-authors (Sokolov et al. 2016) with passive microwave (AMSR-E) previously. We discuss this on lines 395 and 510 of the original manuscript. We fully agree that comparisons to other passive microwave records/frequencies would be useful and specifically exploitation of SSMI/I records in order to be able to go more back in time (see lines 474ff, orig. manuscript), but this is beyond the scope of this study.

> The advantage of using radar is however (1) the potential to analyze the severity of the event (e.g. figure 12c) and (2) the option to go to higher spatial resolution with SAR. SAR is acquired less frequently (with very few exceptions, including Svalbard), so the approach to analyse snow structure change more appropriate. Current SAR missions

do not allow for consistent circumpolar retrieval yet, but it might become feasible in the future.

The intention of the study is to identify advantages and disadvantages of C-band radar. We have now added in the objectives paragraph for clarification: Specifically, the advantages and disadvantages of using C-band radar are assessed.

Note that in a preceding study results from fusion of Ku-band backscatter change with AMSR-E detection are documented (Semmens et al. (2013). It was found that many events detected over Alaska were due to fog instead of ROS: "... fog occurrence is viewed as a proxy for warm air mass intrusion which creates condensation on the snow surface resulting in melt that is detected by the passive microwave" (Semmens et al. 2013, page 9). We briefly discuss this on lines 475-477 of the original manuscript.

Regarding the suggested considerations of Dolant et al. (2016) and Pan et al. (2018): Note that I contributed a review of ROS retrieval methods in Serreze et al. (2021), Table 1, which also provides some basic details of the two papers. Pan et al. is commented there also on page 12, left column. Dolant et al. focused for validation (community observations of ROS from three settlements close to each other) on one winter, 2010/11, which is excluded from our ASCAT/SMOS analyses due to issues in the SMOS records at the beginning of the mission. Pan et al. focus on Alaska and combine passive microwave observations with MODIS to identify if snow is on ground or not. Validation was based on precipitation proxies. These differences in validation strategy across existing studies have been now added to the introduction and discussion. Pan et al.(2018) list several community observations for Fairbanks. A comparison to the final results is not possible as it is too much south and the record has gaps in SMOS, but the unmasked detections have been compared. All 8 events have been detected but usually the day before assigned. We now include this in the discussion on window size.

See also new table 4 which details evaluation strategies in previous studies.

RC1: The combined method of C and L band is interesting. I think the focus of the paper could be narrowed, it is a bit broad and makes the paper harder to understand. I would aim to present the method, then evaluate and not try to make a broad-scale statement on ROS event since it is not properly evaluated yet. Overall, I think this article should be published once modifications are made.

Reply: we have now extended the evaluation using >60 additional snowpit sites. We have included an overview on evaluation methods in previous papers in the discussion to show that our approach even goes beyond on how it is usually done.

References

Dolant, C., Langlois, A., Montpetit, B., Brucker, L., Roy, A., & Roy, A. (2016) Development of a rain-on-snow detection algorithm using passive microwave radiometry. Hydrological Processes, 30, 3184-3196. DOI: 10.1002/hyp.1082

Pan, C. G, Kirchner, P. B, Kimball, J. S, Kim, Y. & Du, J. (2018). Environ. Res. Lett. 13 075004, DOI : 10.1088/1748-9326/aac9d3

Serreze, M. C., Gustafson, J., Barrett, A. P., Druckenmiller, M. L., 660 Fox, S., Voveris, J., Stroeve, J., Sheffield, B., Forbes, B. C., Rasmus, S., Laptander, R., Brook, M., Brubaker, M., Temte, J., McCrystall, M. R., and Bartsch, A.: Arctic rain on snow events: bridging observations to understand environmental and livelihood impacts, Environmental Research Letters, 16, 105 009, https://doi.org/10.1088/1748-9326/ac269b, 2021

Semmens, K. A., Ramage, J., Bartsch, A., and Liston, G. E.: Early snowmelt events: detection, distribution, and significance in a major sub-arctic watershed, Environmental Research Letters, 8, 014 020, https://doi.org/10.1088/1748-9326/8/1/014020, 2013.

Please find our further responses to the 'specific comments' below.

L35. I would not use the term "*aging of snow*" _since it is not accurate. Please refer to the vapour flux from temperature gradient.

- Reply: ‚aging of snow' has been removed

L38. "The mapping of snow changes afterwards instead of wet snow circumvents", I'm not sure what is meant in the sentence. Please consider modification for improved clarity.

- Reply: we suggest the following rephrasing
- Old: The mapping of snow changes afterwards instead of wet snow circumvents this issue but requires the use of wavelengths which are sensitive to changes in snow properties, this means comparably short wavelengths with respect to the typical grain size of snow
- New: The mapping of snow structure changes as a result of events instead of wet snow during an event circumvents this issue but requires the use of wavelengths which are sensitive to changes in snow properties (e.g. Tsang et al. 2022), this means comparably short wavelengths.

L40. Include citation on wavelength and snow grain size.

- Reply: the sentenced has been rephrased and a citation included. See above

l65. Do you mean "W*ith ROS, associated*" _

- Reply: yes

L77-80. Perhaps the objective should be modify or addressed more clearly in the conclusion. Were you able to correctly answer (1) with this method? How was (3) evaluated?

- Reply: "(1) gain insight into recent occurrence of rain on snow events across the Arctic" – this refers to the ROS cases with known impact which are detailed in the paper. We added now 'specific' before 'rain'. Regarding (3), the impact of ROS on snow properties was investigated using hardness from snow pit records. This is briefly

referred to on line 522 in the conclusions, but we agree that it could be extended, also considering the additional results. More than 60 new snow pits were included.

L150. Consider adding a statement on how these data can be subjective and what was done to avoid this.

- Reply: e.g. "Hardness measurements can be subjective. Specific schemes have been developed to judge hardness (see table 1). For long-term measurement sites such as Varanger, Saariselkä and Sodankylä people doing the measurements undergo training in using these schemes. "

L207. Please clarify this sentence "ROS using wet snow from C-band". Do you mean wet snow detection?

- Reply: yes. Rephrasing suggestion: "ROS identification based on wet snow detection from C-band …"

L.246. Please reword the beginning of the sentence.

- Reply: please find our suggestion below
- Old: Passive microwave observations as available from SMOS provide two polarizations ..
- New: Passive microwave observations commonly provide two polarizations ...

L248. Algorithms for ROS detection using 37 and 19 GHz are also sensitive to dry snow surface change into ice crust and ice layer. Consider using does to improve the algorithm since L band is useless when no liquid water is present.

- Reply: In this section we list published wet snow detection schemes. L-band is only used for wet snow in our study, to complement C-band radar. But we agree, that this should be mentioned in the outlook when referring to potential use of other passive microwave data . We have modified both sections. See tracked changes version.

L521-523 Can you provide a quantitative validation of the method to detect ROS events?

- Reply: we have now added more snowpit sites (62 new) to the analyses to allow quantitative assessment. We have in addition made an overview on validation approaches in previous studies (table in discussion) and discuss our results compared to previous studies.

L527. the phrasing with the comma is confusing, do you mean ... "*play a role on what should be considered*"

- Reply: We mean that the role of frequency and polarization should be studied in more detail.
- New: The role of polarization as well as the frequency/wavelength need to be studied in more detail.

L526. Maybe consider using a passive observation with 19 and 37 GHz to improve sensitivity to ice crust and dry snow surface change. Once the liquid water is frozen and the temporal

timing of the ROS event could not be detected with SMOS, those frequencies could help to detect surface change while C-band can provide info at high resolution.

- Reply: see response to 'L248'

L529. "*The magnitude of specific extreme events can be documented by the use of ASCAT alone, without fusion with SMOS.*" I thought you showed you need wet snow detection and ASCAT alone cannot detect ROS.

- Reply: what we mean is that if it was known from other observations that it was a ROS situation, then the cross-check that it was not a 'temperature-drop' misclassification is not needed. It refers to the Alaska example, where the ROS occurred rather south, where there was a gap in SMOS.

Figure 6. This figure is hard to understand. what are H4 1, H4 2 and H4 3? Why not add all layers so we have a better understanding of the whole snowpack?

- Reply: In case of use of 1, 2, and 3, there have been 3 separate layers with type H4 in the snowpack. The figure has been revised and all layer in case of Sodankylä included.

**Reviewer 2**

R2: In this manuscript "Towards long-term records of rain-on-snow events across the Arctic from satellite data" the authors have presented an approach to mapping ROS events by combining observations from C-band scatterometer, which are indicative of snow structure change and L-band passive microwave radar, which is sensitive to the presence of wet snow. The results highlight the added value of using L-band observations to filter out false detections from the ASCAT C-band data which can occur as a result of temperature drops. In addition the authors have made use of additional data sources to support the remote sensing detections, including snow pit data, AWS measurements and caribou data. The use of C-band Sentinel-1 SAR observations was also described and presented, but in my opinion these data played a much smaller role in the data analysis than what I was expecting from earlier on in the manuscript. Overall the method is convincing and the datasets produced are valuable in providing a better understanding of ROS occurrence across the Arctic. However, I have a few general and specific comments which should be straightforward to address before the manuscript is published.

1. Area of study: it is apparent from the figures that the authors have applied the approach to not only the entire Arctic but also to land areas extending much further south (eg., Figure 5). However, it is stated in the conclusion that the approach is only recommended for regions north of 66 deg due to coverage issues with SMOS. Perhaps the authors should revise the boundaries of the areas for which ROS detections are presented, or comment on how representative the ROS data are for the lower latitude areas shown in Fig.5?

   Reply: We are now presenting only results in the time series north of 65°N. The maps still show all 60°N but a line for 65°N has been added in all maps. In addition, an example for data coverage from SMOS for one winter is included which justifies the chosen latitude.

2. The authors have described and presented a wide range of different types of observations, but I think that some of the datasets used do not really add much to the overall goals and conclusions of the study. While I see the need for observations to validate/support the remote sensing data, I think the use of too many different observations, each with their own considerations for ROS detection, makes it at times difficult to follow the main objectives of the study. I would for example recommend reconsidering whether the use of the caribou data are really necessary.

   Reply: The intention of inclusion of the Caribou study is to (1) demonstrate that there is stronger variability from year to year regionally than what can be observed for the entire Arctic (Figure 5) and (2) to point to the potential use of the backscatter change magnitude in addition to just event detection. The Caribou herds did not fully avoid areas where an event was detected, but avoided areas which exceeded a certain value (figure 12c). We have now added text in the discussion part of the paper.

While I think the presentation of Sentinel-1 observations is interesting, I don't think it featured enough in the results to be worth including in the data descriptions/method. Perhaps the authors might consider a follow-up study which focuses primarily on Sentinel-1 instead of including it here?

- Reply: We agree that it would be interesting to do a study with focus on SAR. We have, however, kept the Sentinel-1 analyses as it was also pointed out by reviewer 1

that it adds an interesting aspect. We have added text in the conclusions regarding the SAR results.

Specific comments

Line 175: why were different terms/hardness scales used at Yamal compared to the Scandinavian sites? Why not just use a standard scale for all sites?

Reply: The surveys come from different (partially long-term) monitoring programs, carried out by different institutions which follow different schemes. It would be indeed very beneficial if future surveys would follow the same scheme. We have now decided to drop the hardness records from the Yamal analyses as it is not available from all five points for each sites of the transect. Only the crust description is complete. We therefore limit now the results to the crusts.

Line 296: "location specific threshold" - does this mean that a threshold is determined for individual pixels, or for regions?

Reply: Yes, the threshold is defined individually for each grid point. We now added after location: "(grid point)"

Table 2: Events represent November 2021 to February 2022; why are values from only 1 year/winter of observations used?

Reply: Thanks for spotting! It should read November – February, years 2011-2022

Figure 1: missing?

Reply: The figure is placed further up in the main text.

Figure 2: Please consider splitting into 2 figures as it is very difficult to see the circles showing the reference sites

Reply: The reference sites are also indicated with numbers. We have now increased the figure as well as labels for the reference sites and added them to all panels.

Figure 4: Could the authors comment on the event confirmed by SMOS occurring in the start of December 2016? Here the AWS data show very low temperature (approx. -15 deg.C), no precipitation and increasing snow depth in the following days. What could be the reason for detection of wet snow?

Reply: The SMOS detection actually refers to a smaller event (LRI < 1mm) two days before the ASCAT detection. The ASCAT detection does however represent a period of temperature drop (following the rain event). The impact of liquid precipitation is not captured. This case shows the disadvantage of using a 3 day window for the SMOS masking, but it is necessary to account for data gaps and nature of the radar retrieval scheme, as discussed on lines 304 in the methods part. We agree that the discussion on the choice of the detection window could be more extensive and included in the discussion. We added a comment of RFI impacts with reference to this example and a comparison with event records listed in Pan et al. (2018). Assignment of date can be +- 2 days, what justifies the window size.

Figure 5: Consider splitting into 2 figures; the pan-Arctic maps showing ROS frequency are quite small but show very interesting data. Also consider increasing the symbol size in the legend.

Reply: split as suggested

Figure 8: Legend symbols are far too small. Also move the legend for ASCAT backscatter change closer to the actual figure showing backscatter change (right panel of Fig. 8b)

Reply:revised

Figure 10: Increase the symbol and text sizes in the legend Figure 11: Missing labels (a) and (b)

Reply: labels were inside the maps. revised

Figure 12: Consider removing the caribou observations (panel c) and increase legend/symbol sizes elsewhere

Reply: revised, but caribou study kept, see above

Figure 13: I didn't find that this figure showed any useful information, consider removing it

Reply: see our comment above. We now refer to it in the discussion

Technical corrections

Line 38: change "afterwards" to "following a ROS event" - changed

Line 42: change "ROS events" to "ROS event" - changed

Line 99: Change "production if" to "production of" - changed

Line 129: Define EASE2 - added

Line 130: Define RFI - added

Line 161: Change "University Tromso" to "University of Tromsø" - changed

Line 162: "information is collected" -> "information has been collected" -changed

Line 163: "is used since" -> "has been used since"  -changed

Line 170: "First snow pits" -> "The first snow pits" -changed

Line 181: recommend changing "instrumented" to "equipped"

Line 199: "what is reflected to a certain magnitude depending on" -> "which is reflected to a certain degree by" -changed

Line 202: "is increasing in" -> "increases"  - changed

Line 204: "is significantly decreasing" -> "decreases significantly" -changed

Line 208: "investigated for Norway" -> "investigated for the Svalbard archipelago" - changed

Line 209: "alter" -> "alters" and "even ice layers are forming what allows" -> "the formation of ice layers allows"  - changed

Line 226: "found independent" -> "found to be independent"-changed
Line 232: "real part" -> "the real part"-changed
Line 235: "The attenuation of dry snow layer" -> "The attenuation by dry snow layers" - changed

Line 247: "Band rations" -> "Band ratios"-changed
Line 257: "towards south" -> "towards the south"-changed
Line 300: Figure 4 should be referred to after Figure 3-order changed
Line 306: something missing: "?" – reference adjusted
Line 307: "the finally determined date" -> "the final date determined" – we removed 'finally' as it was used in the meaning of 'eventually'
Line 309: Figure 11 is referred to too early? Figure 5 should come after Figure 4 Line 324: Table ?? - missing number – order changed, reference for Table was supposed to be for Figure 1

Line 331: outlined in ? - missing a section number – refers to a previous publication, now corrected
Line 398: Missing word after "in the following" - winter/summer? – removed, as it is clear to be susequent from 'impacts'
Line 435: "what leads to" -> "which leads to" - changed
Line 439: "In same regions" -> "In the same regions" - changed
Line 440: "Problematic are also" -> "Also problematic are" - changed
Line 441: "change is decreasing" -> "change decreases" - changed
Line 449: "what underlines" -> "which underlines" – changed

Line 464: "Near coastal" -> "Coastal" -changed
Line 481: "Both from Senintel-1" -> "Both Sentinel-1" – word order changed to 'Both commonly available polarization options from Sentinel-1'
Line 482: "as ASCAT" -> "to ASCAT" – changed to 'like ASCAT'
Line 483: "In cases" -> "In some cases" - changed
Line 500: "In case of" -> "In the case of" - changed
Line 507: "what is an issue" -> "which is an issue" - changed
Line 513: "what might explain to" -> "which might explain" - changed
Line 514: "in case of" -> "in the case of" - changed
Line 516: "what corresponds" -> "which corresponds" - changed
Line 527: "what should be" -> "which should be" - changed
Line 533: "what allows" -> "which allows" - changed